

# Conformal field theory on top of a breathing one-dimensional gas of hard core bosons

**Paola Ruggiero[1*], Yannis Brun[2] and Jérome Dubail[2]**

**1** International School for Advanced Studies (SISSA) and INFN,
Via Bonomea 265, 34136 Trieste, Italy
**2** Laboratoire de Physique et Chimie Théoriques, CNRS,
Université de Lorraine, F-54506 Vandoeuvre-les-Nancy, France

⋆ pruggiero@sissa.it

## Abstract

The recent results of [1], which aim at providing access to large scale correlation functions of inhomogeneous critical one-dimensional quantum systems —e.g. a gas of hard core bosons in a trapping potential— are extended to a dynamical situation: a breathing gas in a time-dependent harmonic trap. Hard core bosons in a time-dependent harmonic potential are well known to be exactly solvable, and can thus be used as a benchmark for the approach. An extensive discussion of the approach and of its relation with classical and quantum hydrodynamics in one dimension is given, and new formulas for correlation functions, not easily obtainable by other methods, are derived. In particular, a remarkable formula for the large scale asymptotics of the bosonic $n$-particle function $\langle \Psi^\dagger(x_1, t_1) \ldots \Psi^\dagger(x_n, t_n) \Psi(x'_1, t'_1) \ldots \Psi(x'_n, t'_n) \rangle$ is obtained. Numerical checks of the approach are carried out for the fermionic two-point function —easier to access numerically in the microscopic model than the bosonic one— with perfect agreement.



# 1 Introduction

## 1.1 Context

Tremendous progress has been made in this early XXI$^{\text{st}}$ century in the physics of one-dimensional (1d) quantum systems, both on the experimental and theoretical sides. In particular, it has been realized in the past 15 years that several historical models of many-body quantum physics in 1d, such as the Lieb-Liniger model or the XXZ spin chain, that were previously regarded as oversimplified toy-models, were in fact very good descriptions of real systems that can be created and manipulated in the laboratory [2–13] . In the past two years, we have witnessed an important breakthrough with the development of a "Generalized HydroDynamic" description [14,15] of these systems (see also Refs. [16–30] for further developments) which, contrary to previously existing hydrodynamic approaches, is able to reproduce experimental observations of out-of-equilibrium isolated integrable quantum systems [31–33]. Although it applies to systems that are definitely made of quantum objects (cold atoms), the hydrodynamic approach leads to a classical description of the system.

At the moment an important open problem is to understand how to extend the new "Generalized HydroDynamic" approach to incorporate quantum effects, such as quantum correlations or interference effects. This is the motivation for the present work. We shall focus on a particular concrete problem, which is simple enough so that it can be treated entirely analytically.

While it is known that hydrodynamics accurately predicts one-point functions but fails instead to predict higher-point functions (more specifically, it simply predicts them to be zero),

here we are going to explain how it can be extended to obtain more generic quantum correlation functions. The idea is to use the classical hydrodynamics solution as the background on which one can build an effective action for the quantum fluctuations.

The problem we will treat is the one of a gas of hard-core bosons, also known as the Tonks-Girardeau gas, in a time-dependent harmonic potential $V(x, t)$. This problem is well-known to be exactly solvable [34–36], and our goal is to use it to illustrate our approach, which extends recent works by others and by ourselves and our collaborators [1,37–51]. We will see that we recover some known results about equal-time correlations, and we uncover new ones, including results for correlation functions at different time.

## 1.2 The model, and goal of the paper

The time-dependent Hamiltonian of 1d bosons with delta repulsion in an external potential $V(x, t)$ is

$$H(t) = \int dx \left( \frac{\hbar^2}{2m} (\partial_x \Psi^\dagger)(\partial_x \Psi) + V(x, t)\Psi^\dagger \Psi + g\Psi^{\dagger 2}\Psi^2 \right), \tag{1}$$

where $\Psi^\dagger(x)$, $\Psi(x)$ are operators that create/annihilate a boson at position $x$, which satisfy the canonical commutation rule $[\Psi(x), \Psi^\dagger(x')] = \delta(x - x')$. In this paper we focus exclusively on the hard-core limit (or Tonks-Girardeau limit),

$$g \to +\infty, \tag{2}$$

and on a harmonic trapping potential with a time-dependent frequency,

$$V(x, t) = \frac{1}{2} m \omega(t)^2 x^2. \tag{3}$$

At $t = 0$, the system is in the ground state $|\psi_0\rangle$ of $H$ with an initial trap frequency $\omega_0$ and a chemical potential $\mu$. It is well-known that, in 1d, hard-core bosons can be mapped to free fermions. Using this, the number of bosons $N$ in the ground state is easily calculated: it is equal to (the integer part of) $\mu/(\hbar\omega_0)$. Importantly, this means that

$$\hbar N = \mathcal{O}(\mu/\omega_0), \tag{4}$$

so when we work in units where $\mu$ and $\omega_0$ are both of order 1, taking $N$ very large is equivalent to taking $\hbar$ very small.

In that sense, for this problem, *the thermodynamic limit is a semiclassical limit*. To keep track of quantumness in the problem, we do not set $\hbar$ to one; instead, we will sometimes set

$$\omega_0 = \mu = m = 1. \tag{5}$$

Our goal is to learn how to calculate correlation functions of local observables,

$$\langle O_1(x_1, t_1)O_2(x_2, t_2)\dots O_p(x_p, t_p)\rangle, \tag{6}$$

in the limit $1/N \sim \hbar \to 0$. Throughout the paper, $\langle . \rangle$ is the expectation value in the initial state $|\psi_0\rangle$.

A number of results on this particular problem are available in the literature. These include exact finite-$N$ results that exploit the mapping to fermions for correlation functions of some observables *at equal time* $t_1 = t_2 = \dots = t_p$ [34] (see also Refs. [52–59]); however to our knowledge, such results do not exist for correlations at different times. There are also results for observables that are not straightforwardly expressed in terms of the underlying free fermions and that have been derived in the thermodynamic/semiclassical limit above,

for the *static case*. These include the one-particle density matrix [60] or the entanglement entropy [61]. We will see below that the approach we take here [1,37–48,50,51] automatically reproduces the large-$N$ asymptotics of these known results; moreover, it gives access to correlations at different times.

The manuscript is organized as follows. In section 2 we present the approach we follow throughout the paper, starting with the Wigner function of the free fermion problem, the reduction to a classical hydrodynamic description, and the reconstruction of quantum fluctuations and correlations on top of that classical description. In section 3 we introduce a few notations and useful formulas that are specific to the problem of the time-dependent harmonic oscillator; those formulas follow naturally from a "holographic" picture [62] which we briefly review. In section 4 we use that formalism to write the asymptotics of correlation functions of boson creation/annihilation operators, see formula (52). In particular, this yields a remarkably simple formula for the $n$-particle correlation function at equal time $\langle\Psi^{\dagger}(x_1,t)\dots\Psi^{\dagger}(x_n,t)\Psi(x_1',t)\dots\Psi(x_n',t)\rangle$, see formula (53). In section 5 we focus on the fermionic two-point function. We evaluate its asymptotics using our approach —including low-energy contributions captured by the CFT, but also an important contribution that lies beyond CFT, which we properly take into account— and compare to a numerical evaluation of that two-point function at finite $N$. We find perfect agreement, thus validating the approach. We conclude in section 6.

# 2 Strategy: reconstruction of quantum fluctuations on top of a classical hydrodynamic solution

In this section we outline the general principles that underly the approach we take in this paper, without focusing on any specific correlation function. We briefly sketch the most important ideas of the classical hydrodynamic description of the gas, and of its quantization. Concrete examples of calculations of correlation functions are given in the next sections.

We stress that the reconstruction of quantum fluctuations on top of a classical hydrodynamic solution is an old problem in theoretical physics, dating back to Landau [63], with contributions by many authors motivated by different topics [64–66]. An inspiring treatment is given by Abanov in Ref. [37], which we partially follow here.

## 2.1 Generalized Hydrodynamics/Wigner function evolution

It is well known that hard core bosons in 1d are equivalent to free fermions. The mapping from bosons to fermions is done by inserting a Jordan-Wigner string,

$$\Psi_F^{\dagger}(x) = e^{i\pi\int_{y<x}\rho(y)dy}\,\Psi^{\dagger}(x),\tag{7}$$

where $\rho(y) = \Psi^{\dagger}(y)\Psi(y) = \Psi_F^{\dagger}(y)\Psi_F(y)$ is the density operator, such that the fermion creation/annihilation modes satisfy the canonical anti-commutation relations $\{\Psi_F(x),\Psi_F^{\dagger}(x')\} = \delta(x-x')$. In terms of the fermions, the Hamiltonian (1) in the hard core limit is quadratic,

$$H(t) = \int dx\left(\frac{\hbar^2}{2m}(\partial_x\Psi_F^{\dagger})(\partial_x\Psi_F) + V(x,t)\Psi_F^{\dagger}\Psi_F\right).\tag{8}$$

To go towards a hydrodynamic description, it is useful to introduce the Wigner function of these free fermions,

$$n(x,k,t) = \int dy\,e^{iky}\left\langle\Psi_F^{\dagger}(x+y/2,t)\Psi_F(x-y/2,t)\right\rangle,\tag{9}$$

which has the semiclassical interpretation of being the probability to find a fermion at position $(x, k)$ in phase space. The Wigner function satisfies the exact evolution equation

$$\partial_t n(x, k, t) + \frac{\hbar k}{m} \partial_x n(x, k, t) = \frac{1}{\hbar}(\partial_x V(x, t))\partial_k n(x, k, t). \tag{10}$$

We stress that this equation is exact because, in this paper, we focus exclusively on harmonic potentials $V(x, t)$. For more general potentials, Eq. (10) would be the leading order term in an expansion in $\hbar$ and in higher order derivatives $\partial_x^p V$, $p \geq 2$ [67].

Eq. (10), which holds for free fermions, is the simplest possible occurence of the Generalized HydroDynamics equations (GHD) discovered in 2016 [14,15]. It would be very tempting to try to generalize what we are doing here to the interacting case, replacing Eq. (10) by GHD; we plan to come back to this in subsequent papers. Here we explain the basic ideas by focusing on the simplest case which is entirely analytically solvable: the Tonks-Girardeau gas in a harmonic trap.

What is important is that Eq. (10) is a classical evolution equation, so the quantumness of the problem must be entirely hidden in the initial condition $n(x, k, t = 0)$. We make the problem even more classical by considering a particular class of *classical* or *zero-entropy* initial states (in the terminology of [17,69]), which then leads to an entirely classical description of the problem. Quantum fluctuations will be reconstructed later, by re-quantizing the resulting classical hydrodynamic description.

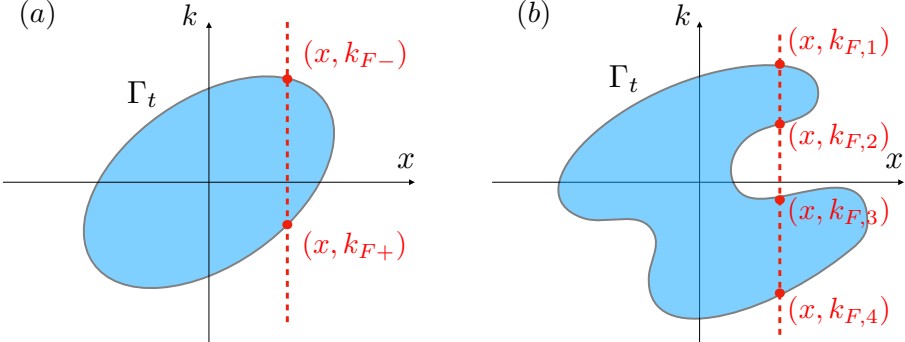

Figure 1: The curve $\Gamma_t$ encircling the points $(x, k)$ at which the Wigner function $n(x, k, t)$ is equal to one. Panel (a): the simple situation considered in this paper, where at any given position $x$ there are only two Fermi points on the contour $\Gamma$, labeled as $k_{F\pm}$. Panel (b): the general situation with more than two Fermi points, not considered in this paper.

## 2.2 Reduction to classical hydrodynamics

In this paper, we focus on the evolution from the ground state, which is a zero-entropy state. This class of zero-entropy states has been studied in several references [17,68,70,71]. What is important is that in the thermodynamic/semiclassical limit $1/N \sim \hbar \to 0$, the Wigner function $n(x, k, t)$ of the ground state is just an indicatrix function, parametrized by a certain curve $\Gamma_t$ in phase space:

$$n(x, k, t) = \begin{cases} 1, & \text{if } (x, k) \text{ is inside the contour } \Gamma_t \\ 0, & \text{if } (x, k) \text{ is outside } \Gamma_t. \end{cases} \tag{11}$$

The evolution equation (10) for the Wigner function can be viewed as an evolution equation for the Fermi contour $\Gamma_t$. One can parametrize $\Gamma_t$ locally as $(x, k_{F,j}(x, t))$ where $k_{F,j}(x, t)$,

$j = 1, 2, \dots$ are the Fermi points at position $x$, where the Wigner function jumps from 0 to 1 (see Fig. 1). Eq. (10) is then an evolution equation for the Fermi points which takes the form of a Burgers equation [37],

$$\hbar \partial_t k_{F,j}(x,t) + \frac{\hbar^2 k_{F,j}(x,t)}{m} \partial_x k_{F,j}(x,t) = -\partial_x V(x,t). \tag{12}$$

Importantly, in this paper we will always be dealing with situations in which the number of position dependent Fermi points is at most two, as illustrated in Fig. 1(a). A situation like the one in Fig. 1(b) will never occur. Situations with more than two Fermi points are also very interesting [17, 68, 70, 71], but are deferred to a subsequent paper.

When there are at most two Fermi points $k_{F,1}(x,t)$ and $k_{F,2}(x,t)$ at each position $x$—we label them as $k_{F-}(x,t)$ and $k_{F+}(x,t)$ in that case—, the gas in the small cell $[x, x + dx]$ is locally in a macrostate which is nothing but the ground state of the homogeneous gas up to a Galilean boost. This simple situation is then captured by the standard Euler hydrodynamic equations for the gas. The latter are expressed in terms of the local density and local velocity (defined as the particle current $j$ divided by the density $\rho$),

$$
\begin{aligned}
\rho(x,t) &= \int \frac{dk}{2\pi} n(x,k,t) = \frac{k_{F-}(x,t) - k_{F+}(x,t)}{2\pi}, \\
u(x,t) &= \frac{j(x,t)}{\rho(x,t)} = \frac{1}{\rho(x,t)} \int \frac{dk}{2\pi} n(x,k,t) \frac{\hbar k}{m} = \frac{\hbar}{m} \frac{k_{F-}(x,t) + k_{F+}(x,t)}{2},
\end{aligned}
\tag{13}
$$

and are obtained straightforwardly from Eq. (12),

$$
\begin{cases}
\partial_t \rho + \partial_x(\rho u) &= 0 \\
\partial_t u + u \partial_x u &= -\frac{1}{m\rho} \partial_x P - \frac{1}{m} \partial_x V.
\end{cases}
\tag{14}
$$

The first equation is the continuity equation for $\rho$ which expresses conservation of mass. The second equation is the Euler equation, with a pressure given by $P(\rho) = \frac{\pi^2 \hbar^2 \rho^3}{3m}$, which is specific to hard core bosons in 1d at zero temperature.

We stress that the reduction to standard hydrodynamics is not specific to the free model we are considering: in fact, it has been shown [17] that for zero-entropy states GHD is equivalent to standard hydrodynamics as long as the number of Fermi points is two (i.e. before the appearance of *shocks* in the solution of the hydrodynamic equations (14)). In our case, this means that, for finite repulsion strength in the bosonic model (1), we would still get a system of the form (14), but with the appropriate pressure, i.e. the pressure of the Lieb-Liniger model at zero temperature.

We also stress that in the initial state, $u = 0$, so the classical hydrodynamic equations (14) simplify to a *hydrostatic* equation: $\frac{1}{m\rho} \partial_x P = -\frac{1}{m} \partial_x V$. Using thermodynamic relations, this equation can be rewritten as $\partial_x[\mu + V(x)] = 0$. This is solved by tuning locally the chemical potential, $\mu \to \mu(x) = \mu - V(x)$, so it is equivalent to the Local Density Approximation (LDA).

At this point, we have reached an entirely classical description of the gas in terms of standard hydrodynamic equations (14). In that description, the "quantumness" resides exclusively in the equation of state—i.e. in the formula for the pressure—.

In particular, at this level of description, the *connected parts of all correlation functions at equal time*—but different positions $x_i \neq x_j$ in Eq. (6)— *vanish*, because by construction all the fluid cells are independent. The are no correlations at equal time in classical hydrodynamics. Our goal is to learn how to reintroduce quantumness in that description, in order to reconstruct such quantum correlations.

## 2.3 Quantum fluctuations around the classical hydrodynamic solution

In order to re-quantize the problem, we start by writing an action for the density $\rho(x,t)$ and the associated current $j(x,t) = \rho(x,t)u(x,t)$,

$$S = \int dx dt \left[ m \frac{j^2}{2\rho} - \varepsilon(\rho) - \rho V \right], \tag{15}$$

where $\varepsilon(\rho) = \frac{\pi^2 \hbar^2 \rho^3}{6m}$ is the ground state energy density, and where the two functions $\rho(x,t)$ and $j(x,t)$ are constrained by the continuity equation $\partial_t \rho + \partial_x j = 0$. Let us check that, with that additional constraint, the action (15) is compatible with the Euler equation —the second equation in (14)—. One way to do that is to represent fluctuations around a given classical configuration $(\rho(x,t), j(x,t))$ which satisfies the continuity equation by a height field $h(x,t)$,

$$\delta\rho(x,t) = \frac{1}{2\pi} \partial_x h(x,t), \qquad \delta j(x,t) = -\frac{1}{2\pi} \partial_t h(x,t), \tag{16}$$

such that $(\rho + \delta\rho, j + \delta j)$ also satisfies the continuity equation. Then it is easy to check that the Euler-Lagrange equation for the height field $h$ yields the Euler equation. Indeed, varying the action to the first order in $\delta\rho, \delta j$ gives (substituting (16) and integrating by parts)

$$\begin{aligned}
\delta S^{(1\text{st order})} &= \int dx dt \left[ m \frac{j \delta j}{\rho} - m \frac{j^2}{2\rho^2} \delta\rho - \delta\rho \partial_\rho \varepsilon - \delta\rho V \right] \\
&= \frac{m}{2\pi} \int dx dt \left[ \partial_t \left( \frac{j}{\rho} \right) + \frac{1}{m} \partial_x \left( \frac{j^2}{2\rho^2} \right) + \partial_x (\partial_\rho \varepsilon + V) \right] h.
\end{aligned}$$

The expression inside the bracket must vanish; using $u = j/\rho$ and the thermodynamic relation $\partial_x(\partial_\rho \varepsilon) = \frac{1}{\rho} \partial_x P$, this gives precisely the second equation in (14). So a classical configuration for the action (15) is indeed a solution of the classical hydrodynamic equations (14), as required.

Around such a classical configuration, one can then expand to second order to get

$$\begin{aligned}
S[h] &\equiv \delta S^{(2\text{nd order})} \\
&= \frac{1}{8\pi} \int dx dt \left[ \frac{m}{\pi\rho} (\partial_t h)^2 + 2 \frac{mj}{\pi\rho^2} (\partial_x h)(\partial_t h) + \left( \frac{mj^2}{\pi\rho^3} - \frac{1}{\pi} \partial_\rho^2 \varepsilon \right)(\partial_x h)^2 \right]. \tag{17}
\end{aligned}$$

This gives an action for the quantum fluctuations around the classical hydrodynamic solution. Of course, there are also higher order terms, but these higher order terms are RG irrelevant in two spacetime dimensions, and we will omit them. Quantum fluctuations are thus captured by a quadratic action, a fact that is very well known from Luttinger liquid theory [72, 73].

## 2.4 CFT in emergent curved spacetime

The next step is to rewrite the quadratic action (17) in a more friendly form,

$$S[h] = \frac{\hbar}{8\pi} \int g^{ab} (\partial_a h)(\partial_b h) \sqrt{-\det g} \, d^2\mathbf{x}, \tag{18}$$

with coordinates $\mathbf{x}^0 = t$ and $\mathbf{x}^1 = x$, and where $g^{ab}$ are the components of the inverse of the $2 \times 2$ matrix

$$g = \begin{pmatrix} \frac{mj^2}{\pi\hbar\rho^3} - \frac{1}{\pi\hbar} \partial_\rho^2 \varepsilon & -\frac{mj}{\pi\hbar\rho^2} \\ -\frac{mj}{\pi\hbar\rho^2} & \frac{m}{\pi\hbar\rho} \end{pmatrix},$$

that we want to interpret as a *metric tensor*. [Notice that, to arrive at Eq. (18), we use the explicit form of the internal energy $\varepsilon(\rho) = \frac{\pi^2 \hbar^2 \rho^3}{6m}$. This result is then specific to the hard core limit of 1d bosons, which map to free fermions. In contrast, when dealing with truly interacting 1d liquids, one would end up with a quadratic action of a slightly more general form than (18), with a position-dependent coupling constant, see Refs. [46, 48] for full details.]

The action (18) is invariant under diffeomorphisms, and also under Weyl transformations of the metric $g_{ab} \to e^{2\sigma} g_{ab}$. The metric can thus be rescaled and be put in the form

$$g \to \frac{\pi \hbar \rho}{m} g = \begin{pmatrix} u^2 - v_F^2 & -u \\ -u & 1 \end{pmatrix},$$

where $v_F(x,t) = \frac{\pi \hbar}{m}\rho(x,t)$ is the Fermi velocity, so that

$$ds^2 = g_{ab}dx^a dx^b = -v_F(x,t)^2 dt^2 + (dx - u(x,t)dt)^2. \qquad (19)$$

Written in this way, the emergent spacetime metric (19) —fixed by the classical hydrodynamic background $\rho(x,t), u(x,t)$— possesses a clear interpretation. The gas consists of local fluid cells, which locally look like the ground state of the translation invariant Tonks-Girardeau gas, up to a Galilean boost. The velocity of sound waves, or of gapless excitations around the ground state at density $\rho$, is the Fermi velocity $v_F = \frac{\pi \hbar}{m}\rho$. In the ground state the metric that would encode the propagation of those excitations would be $ds^2 = -v_F^2 dt^2 + dx^2$. But here the local macrostate of the gas is boosted, in order for the fluid to have a local mean velocity $u \neq 0$. Thus, locally, the sound waves are also boosted by the hydrodynamic velocity $u$, resulting in Eq. (19). Another way to say this is that such gapless excitations propagate along the null geodesics of the metric (19), and those satisfy the simple equation $\frac{d}{dt}x(t) = u(x(t),t) \pm v_F(x(t),t)$, where the $\pm$ sign corresponds to left or right moving trajectories.

What is important now is that we are in a CFT, so all local observables $O(x,t)$ can be decomposed into left (+) and right (−) moving components, of the form $O_+ \otimes O_-$ or as linear combinations thereof. Each chiral component just follows its own null geodesics, and, following these, every operator can be traced back to its original position at time $t = 0$. In particular, focusing on observables of the form $O = O_+ \otimes O_-$, correlation functions can be rewritten as

$$\left\langle O_1(x_1,t_1)O_2(x_1,t_1)\dots O_p(x_p,t_p) \right\rangle \propto$$
$$\propto \left\langle O_{1+}(x_+^0(x_1,t_1),0)O_{1-}(x_-^0(x_1,t_1),0)\dots O_{p+}(x_+^0(x_p,t_p),0)O_{p-}(x_-^0(x_p,t_p),0) \right\rangle, \quad (20)$$

up to some Jacobian factors and where $x_\pm^0(x_i,t_i)$ is the position at initial time of the (right/left) null geodesics passing through $x_i$ at time $t_i$. Note that in flat space $x_\pm^0(x_i,t_i)$ would simply be $x_i \mp t_i$, while here we have something more complicated because spacetime is curved. This is simplified by using isothermal coordinates, see below. Importantly, the r.h.s. of Eq. (20) is a correlation function in the initial state, so it is a correlation function at equilibrium.

## 2.5 Correlations in the initial state

Correlation functions in the initial state were studied in Refs. [1, 39, 40, 46], where the same trick of incorporating most effects of the inhomogeneity into the metric of the CFT was used. To get correlation functions at equilibrium in the ground state of the trapped gas, one works with a metric with Euclidean signature, $ds^2 = dx^2 + v_F^2(x)d\beta^2$, where $\beta$ is now imaginary time, and $v_F(x) = \frac{\pi \hbar}{m}\rho_0(x)$ is the local Fermi velocity in the ground state. Moreover, there always exist *isothermal coordinates* such that the metric takes the form $ds^2 = e^{2\sigma_0}(d\xi^2 + d\beta^2)$. Then a Weyl transformation ($g_{ab}e^{2\sigma_0} \to g_{ab}$) brings the above metric back to the flat one, $ds^2 = d\xi^2 + d\beta^2$. Under that Weyl transformation, primary fields transform as $O \to e^{-\sigma_0 \Delta}O$,

where $\Delta$ is the conformal dimension of $O$. Therefore the (equilibrium) correlator in the r.h.s. of Eq. (20) becomes

$$\left(\prod_{a=1}^{p} e^{\Delta_p \sigma_0}\right)\left\langle O_1(\xi_1,0)O_2(\xi_2,0)\dots O_p(\xi_p,0)\right\rangle_{\text{flat}}, \tag{21}$$

where $\Delta_p$ is the scaling dimension of $O_p$ and the correlation function is evaluated in the flat metric.

## 2.6 Isothermal coordinates

The procedure we just described is simplified by using *isothermal coordinates* for our ermegent 2d spacetime. In these coordinates the metric is of the form

$$d^2s = e^{2\sigma}(-d\tau^2 + d\xi^2), \tag{22}$$

with a conformal factor $e^{2\sigma}$ given by the Jacobian of the transformation $(x,t) \mapsto (\xi,\tau)$. In particular, in these coordinates, doing the backward evolution is trivial (Fig. 2). For the particular case studied in this paper —the breathing Tonks-Girardeau gas— the explicit coordinate system $(\xi,\tau)$ will be given in Sec. 3. Then Eq. (20) becomes

$$\left\langle O_1(x_1,t_1)O_2(x_1,t_1)\dots O_p(x_p,t_p)\right\rangle$$
$$= \left(\prod_{a=1}^{p} e^{\Delta_p \sigma(x_p,t_p)}\right)\left\langle O_1(\xi_1,\tau_1)O_2(\xi_2,\tau_2)\dots O_p(\xi_p,\tau_p)\right\rangle_{\text{flat}} \tag{23}$$

and, in these coordinates, the dynamics is trivial :

$$\left\langle O_1(\xi_1,\tau_1)O_2(\xi_2,\tau_2)\dots O_p(\xi_p,\tau_p)\right\rangle_{\text{flat}}$$
$$= \left\langle O_{1+}(\xi_1+\tau_1)O_{1-}(\xi_1-\tau_1)\dots O_{p+}(\xi_p+\tau_p)O_{p-}(\xi_p-\tau_p)\right\rangle_{\text{flat}}. \tag{24}$$

Here each chiral component is just a function of one of the null coordinates $\xi_\pm = (\xi \pm \tau)$. Notice that, contrary to Eq. (20), there is no multiplication by Jacobian factors in Eq. (24), because the Jacobians of the transformation $(\xi,\tau) \mapsto (\xi-\tau,\xi+\tau)$ are trivial.

Again, we stress that the correlator in the r.h.s. of (24) is an equilibrium one. Moreover, it is further simplified by the fact that right and left moving components are related in a simple way at the edges of the system, as we now explain.

## 2.7 Reflection against boundaries and "chiralization" of the CFT

At the edges of the cloud, when $1/N \sim \hbar \to 0$, the density vanishes. In this section we argue that, in such limit, from the point of view of the CFT that describes low-energy fluctuations around the classical solution, the edges of the cloud become particularly simple: they are simply encoded as a Dirichlet boundary condition at the left and right edges $x_L(t)$ and $x_R(t)$,

$$h(x_L(t),t) = h(x_R(t),t) = 0. \tag{25}$$

This follows directly from the definition of the height field (16): since the velocity of the right interface is $\dot{x}_R(t) = u(x_R(t),t) = j(x_R(t),t)/\rho(x_R(t),t)$, one gets

$$\frac{1}{2\pi}\frac{d}{dt}h(x_R(t),t) = u_R(t)\frac{1}{2\pi}\partial_x h(x_R(t),t) + \frac{1}{2\pi}\partial_t h(x_R(t),t)$$
$$= u_R(t)\rho(x,t) - j(x_R(t),t) = 0.$$

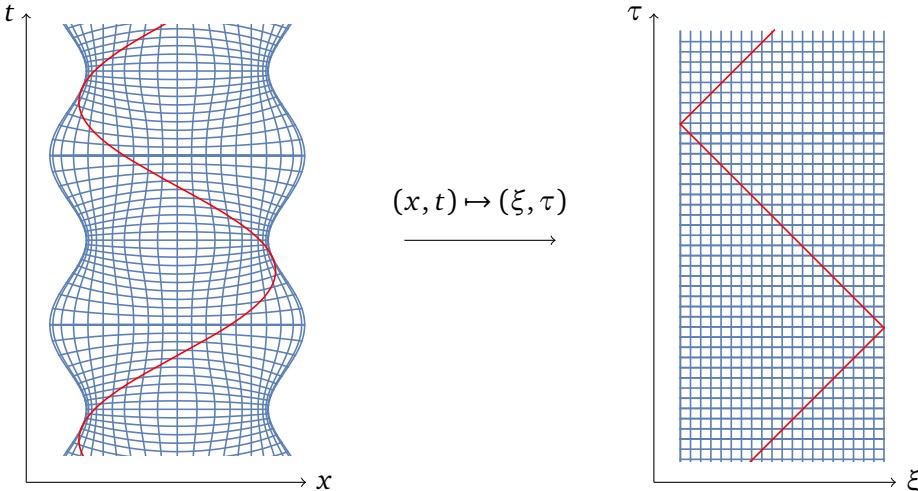

Figure 2: (Left) Spacetime trajectory of a gapless excitation on top of the breathing gas —drawn for the particular case $\omega(t) = \omega_0$ if $t \leq 0$ and $\omega(t) = \omega_1$ if $t > 0$ here—. (Right) In isothermal coordinates $(\xi, \tau)$, the trajectory is simply given by $\xi \pm \tau = $ const. Since we have a conformal boundary condition at the left and right edges, the excitations are simply reflected at the boundaries. This allows to easily trace back observables at position $x$ at time $t$ to their original position at $t = 0$.

Thus $h(x_R(t), t)$ is a constant; the same conclusion holds for $h(x_L(t), t)$. To see that the two constants for the left and right edges are equal, one uses the definition (16) and the fact that the total number of particles $N$ is a constant, so the fluctuations of the density $\delta\rho$ vanish when they are integrated over the whole system:

$$h(x_R(t), t) - h(x_L(t), t) = 2\pi \int_{x_L(t)}^{x_R(t)} dx\, \delta\rho(x, t) = 0.$$

Thus we have $h(x_L(t), t) = h(x_R(t), t) = $ const., and the value of the constant can be chosen arbitrarily; we chose to fix it to zero. We then arrive at the boundary condition (25) as claimed.

At this point the reader might be worried by the fact that the variation of the density is not small at the edge, so the hydrodynamic description might perhaps break down there. We stress that this is not the case. At least, not for our purposes. The hydrodynamic description remains valid arbitrarily close to the edge in the limit $1/N \sim \hbar \to 0$ because separation of scales, which is the key assumption underlying the hydrodynamic approach, holds up to a distance $\delta \sim (\mu m^3 \omega_0^2)^{-1/6} \hbar^{2/3}$ from the boundary. [This is because at a small distance $\epsilon$ from the edge, the density goes as $\sqrt{\epsilon}/\hbar$, so the typical length of density variation is $\sim (\partial_\epsilon \rho/\rho)^{-1} \sim \epsilon$ and the interparticle distance scales as $\sim \hbar/\sqrt{\epsilon}$; separation of scales $\hbar/\sqrt{\epsilon} \ll \epsilon$ holds as long as $\epsilon$ is large compared to $\hbar^{2/3}$.] Thus, at finite $1/N \sim \hbar$, hydrodynamics is valid everywhere except in a small region of width $\delta$ near the edge, and the width of that region goes quickly to zero in the limit $1/N \sim \hbar \to 0$. So we do not have to worry about it, at least not at the larger scales in which we are interested when we look at correlation functions (6).

Now let us come back to the Dirichlet boundary condition (25) and its implications. Since this is a conformal boundary condition, the low-energy excitations are simply reflected against the edge. Left moving operators and right moving ones are just the analytic continuations of one another, as usual in boundary CFT [74]. It is easier to explain this in coordinates $(\xi, \tau)$.

We will see below that the coordinate system $(\xi, \tau)$ can be chosen such that the left and right boundary are at $\xi = -\pi$ and $\xi = 0$. Then it is convenient to think of the height field as being a sum of right- and left-moving components $h(\xi, \tau) = \varphi_-(\xi - \tau) + \varphi_+(\xi + \tau)$, where both

components are constrained by the boundary condition (25). The constraint is implemented by imagining that one has a single right-moving chiral boson $\varphi(\xi - \tau)$ on a circle of circumference $2\pi$, $\varphi(\xi - \tau) = \varphi(\xi - \tau + 2\pi)$, and that

$$h(\xi, \tau) = \varphi(\xi - \tau) - \varphi(-\xi - \tau). \tag{26}$$

Thus $h$ automatically vanishes at $\xi = -\pi$ and $\xi = 0 \pmod{2\pi}$.

   This representation in terms of a single chiral boson is very useful because it allows to trace observables at time $t$ back to the ones at time 0 in a straightforward way. Indeed, right-moving components of observables $O_-(\xi - \tau)$ will be expressible in terms of $\varphi(\xi - \tau)$ only, in some form which we write generically as $O_-[\varphi(\xi - \tau)]$. Similarly, the left-moving component $O_+(\xi - \tau)$ will be expressible as $O_+[\varphi(-\xi - \tau)]$. Then the correlation function (24) becomes a correlator that involves the chiral boson $\varphi$ at different points on the circle $\mathbb{R}/(2\pi\mathbb{Z})$, i.e.,

$$\begin{aligned}
&\big\langle O_1(\xi_1, \tau_1) O_2(\xi_2, \tau_2) \ldots O_p(\xi_p, \tau_p) \big\rangle \\
&= \big\langle O_{1+}[\varphi(-\xi_1 - \tau_1)] O_{1-}[\varphi(\xi_1 - \tau_1)] \ldots O_{p+}[\varphi(\xi_p + \tau_p)] O_{p-}[\varphi(\xi_p - \tau_p)] \big\rangle. 
\end{aligned} \tag{27}$$

The propagator of the chiral boson $\varphi(\xi)$ with $\xi \in \mathbb{R}/2\pi\mathbb{Z}$ in the r.h.s. of Eq. (27) is the one on an infinitely long cylinder (adding the imaginary time direction) with Euclidean metric:

$$\langle \varphi(\xi_1) \varphi(\xi_2) \rangle = -\log \left[ 2 \sin \frac{\xi_1 - \xi_2}{2} \right]. \tag{28}$$

Then all correlation functions of the form (27) can be evaluated by using Wick's theorem.

## 2.8 Summary

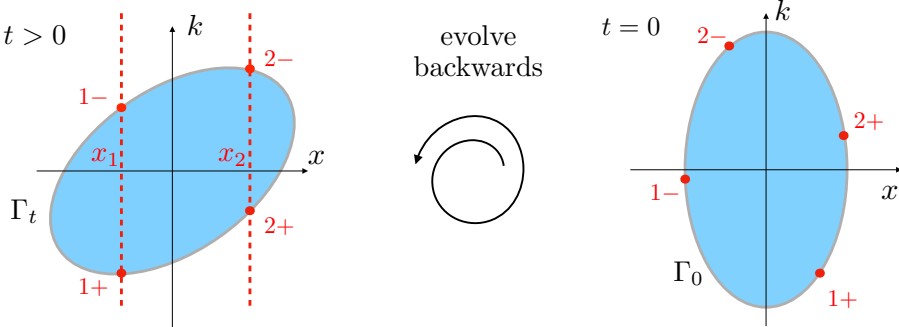

Figure 3: Summary of the approach taken in this paper. To calculate the correlation function $\langle O_1(x_1, t) O_2(x_2, t) \rangle$ at time $t$, we start by writing it in terms of chiral operators $\langle O_{1+} O_{1-} O_{2+} O_{2-} \rangle$ that generate low-energy excitations around the Fermi points $(x_1, k_{F-}(x_1, t))$, $(x_1, k_{F+}(x_1, t))$, $(x_2, k_{F-}(x_2, t))$, $(x_2, k_{F+}(x_2, t))$. Then we trace the Fermi points back to their original positions in phase space. The problem boils down to calculating a correlation function of observables along the initial Fermi contour $\Gamma_0$.

   In the end, we have reached the following simple framework to compute a correlation function of the form (6). First of all, since we are in CFT, every operator $O_i$ can be decomposed in chiral components acting on low-energy fluctuations around the Fermi points $(x_i, k_{F\pm}(x_i, t))$ for $i = 1, \cdots, p$. Those Fermi points are identified in phase space as the intersections of the vertical line passing through $x_j$ and the Fermi contour $\Gamma_t$, see Figs. 1 and 3. Then we "play the movie backwards" in phase space, i.e. we trace the Fermi points back to their original positions in phase space. Actually, the whole contour $\Gamma_t$ can be traced back to its initial configuration $\Gamma_0$

and the problem boils down to calculating a correlation function of chiral observables along the contour $\Gamma_0$ at time $t = 0$.

The general procedure, however, can be simplified if one is able to find a set of *isothermal* coordinates $(\xi, \tau)$ (Eq. (22)) and, in this case, the recipe to perform the calculation is as follows. For each $(x_j, t_j)$ one first needs to identify the corresponding $(\xi_j, \tau_j)$. By performing the associated Weyl transformation, we are left with the same correlator but in the usual CFT in flat spacetime (Eq. (23)). Then one simply traces each chiral component of the observable sitting at $(\xi_j, \tau_j)$ back to its initial position at time zero, i.e., $(\xi_{j,\pm}, 0)$, with $\xi_{j,+} = \xi_j - \tau_j$ and $\xi_{j,-} = \xi_j + \tau_j$ for right and left movers respectively (Eq. (24)), ending up with the problem of computing a correlation function of (chiral) observables at equilbrium. Because of the simple Dirichlet boundary condition that we found at the edges of the system, left and right movers are simply reflected against the boundaries, and correlation functions can be expressed in terms of a single chiral bosonic field $\varphi$ living on a circle of circumference $2\pi$ (Eq. (27)).

All those steps lead to a simple formula in the end,

$$
\begin{aligned}
&\left\langle O_1(x_1, t_1) O_2(x_1, t_1) \ldots O_p(x_p, t_p) \right\rangle \\
&= \left( \prod_{a=1}^{p} e^{\Delta_p \sigma} \right) \left\langle O_{1+}[\varphi(-\xi_1 - \tau_1)] O_{1-}[\varphi(\xi_1 - \tau_1)] \ldots O_{p+}[\varphi(\xi_p + \tau_p)] O_{p-}[\varphi(\xi_p - \tau_p)] \right\rangle,
\end{aligned}
\tag{29}
$$

where the r.h.s. is understood as a correlator in the ground state of a CFT in Euclidean spacetime and can be evaluated using only the propagator of the bosonic field $\varphi$ (Eq. (28)) and Wick's theorem.

# 3 Time-dependent harmonic trap: useful formulas

In this section we provide the explicit formulas that solve the classical hydrodynamic equations (14) for the time-dependent harmonic trap, and give the corresponding isothermal coordinates $(\xi, \tau)$ that allow to propagate correlation functions at arbitrary times back to the ones in the initial state, see Eq. (24). This is greatly simplified by the existence of a "holographic" picture for the time-dependent harmonic potential [62], which we briefly describe.

## 3.1 A simple "holographic" model: 2d picture of the 1d time-dependent harmonic oscillator (after Ref. [62])

Consider a classical rigid pendulum of length $\ell(t)$ attached to the origin in the two-dimensional plane $(x, y)$, and rotating freely around it. The position of the endpoint of the pendulum is parametrized as

$$
\begin{cases}
x(t) = \ell(t) \cos \xi(t) \\
y(t) = \ell(t) \sin \xi(t),
\end{cases}
\tag{30}
$$

where $\xi(t)$ is defined modulo $2\pi$. At time $t = 0$, the length of the pendulum is $\ell_0$ and its angular velocity is $\omega_0$. The length $\ell(t)$ varies, so the energy of the pendulum is not conserved, but its angular momentum is. This implies

$$
\frac{d\xi(t)}{dt} = \frac{\omega_0}{b(t)^2},
\tag{31}
$$

where $b(t) = \ell(t)/\ell_0$. Now, we imagine that for some reason we are unaware of the two-dimensional nature of the problem, and that we have access only to the projection of the

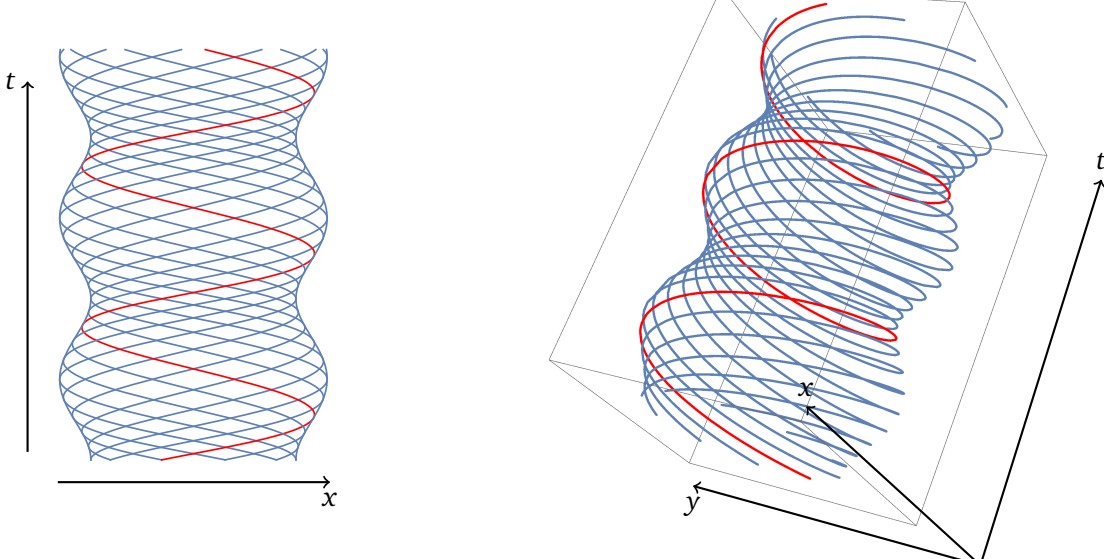

Figure 4: (Left) trajectories $x(t)$ of a classical pointlike particle in a harmonic trap with time-dependent frequency (here plotted for the particular choice $b(t) = \sqrt{1 + (\omega_0^2/\omega_1^2 - 1)\sin^2(\omega_1 t)}$ corresponding to a sudden quench of the frequency $\omega_0 \to \omega_1$ at $t = 0$). (Right) "Holographic" picture [62]: the trajectories $x(t)$ can be viewed as the projection of the 2d motion of a pendulum rotating around the origin in the $x - y$-plane, thus making the conservation of 2d angular momentum (31) obvious.

motion along the $x$-direction. We observe the trajectory $x(t)$, and what we see is precisely the motion of a point-like particle *in a time-dependent harmonic potential* [62],

$$m\frac{d^2x}{dt^2} = -\partial_x V, \qquad \text{where} \qquad V(x,t) = \frac{1}{2}m\left(\frac{\omega_0^2}{b(t)^4} - \frac{\ddot{b}(t)}{b(t)}\right)x^2. \tag{32}$$

[This follows straightforwardly from plugging $x = \ell\cos\xi$ into Newton's equation, using conservation of angular momentum (31).] Thus, from now on, we assume that the function $b(t)$ is related to the time-dependent frequency of our trap (3) through the differential equation

$$\ddot{b}(t) + \omega(t)^2 b(t) = \frac{\omega_0^2}{b(t)^3}, \tag{33}$$

known as the *Ermakov equation* [75,76], with the initial condition $b(0) = 1$ and $\dot{b}(0) = 0$. [The latter condition is imposed because in our problem we are assuming that we are initially at rest in a trap with frequency $\omega_0$, so $\dot{b}(t) = 0$ for $t < 0$, and $\dot{b}(t)$ must be continuous in order to be a solution of (33).] From this higher-dimensional perspective, and assuming that the function $b(t)$ has been calculated for the given $\omega(t)$ defining our problem —see Ref. [76] for the solution to Eq. (33)—, it is straightforward to calculate the trajectory of the classical pointlike particle in the time-dependent trap: it is simply given by $x(t) = \ell_0 b(t)\cos(\xi(0) + \tau(t))$ with

$$\tau(t) \equiv \int_0^t \frac{\omega_0\,ds}{b(s)^2}. \tag{34}$$

As we will see shortly, the trajectories $(x(t), t)$ are nothing but the null geodesics of the metric (19), and the coordinates $(\xi, \tau)$ are isothermal for that metric.

## 3.2 Formulas and parametrizations

This 2d picture of the 1d time-dependent harmonic oscillator leads to simple explicit formulas that are useful in the context of the formalism of Sec. 2. We start by writing the $1/N \sim \hbar \to 0$ limit of the Wigner function in the form (11). In the initial state, it is obtained by saying that all points $(x, k)$ in phase space are occupied iff their total energy $E(x, k) = \frac{\hbar^2 k^2}{2m} - \mu + \frac{m\omega_0^2 x^2}{2}$ is negative. Thus, the curve $\Gamma_{t=0}$ that encloses all these points is an ellipse. Then, at $t > 0$, since the motion of each semi-classical particle is given by $x(t) = b(t)\cos(\xi(0) + \tau(t))$, its phase space position is $(x(t), k(t)) = (x(t), \frac{m}{\hbar}\frac{dx(t)}{dt})$. Hence, the curve $\Gamma_t$ of Eq. (11) is the (rotated) ellipse

$$(x, k) \in \Gamma_t \quad \Longleftrightarrow \quad \frac{b(t)^2(\hbar k - m x \dot{b}(t)/b(t))^2}{2m} - \mu + \frac{m\omega_0^2 x^2}{2b(t)^2} = 0. \tag{35}$$

**Classical hydrodynamic solution.** Using Eqs. (13) one gets the following solution to the classical hydrodynamic equations (14),

$$\rho(x, t) = \frac{1}{b(t)} \times \frac{m\omega_0}{\pi\hbar}\sqrt{\frac{2\mu}{m\omega_0^2} - \frac{x^2}{b(t)^2}}, \qquad u(x, t) = x\,\frac{d\log b(t)}{dt}. \tag{36}$$

This explicit solution is well known in the literature and is usually referred to as a *scaling solution* [34–36, 77] since the density profile at time $t$ is a simple rescaling of the one at time zero.

**Isothermal coordinates** $(\xi, \tau)$. Next, we turn to the metric (19). By construction, the null geodesics of that metric are the trajectories of low-energy excitations around the classical solution (36), or in other words of points in phase space that lie along the Fermi contour $\Gamma_t$. Thus the geodesics read simply $x(t) = b(t)\cos(\xi + \tau)$, and we see, as anticipated by our notations, that the coordinate system $(\xi, \tau)$ is indeed isothermal for the metric (19). To be more precise, for each $x \in [-b(t)\sqrt{\frac{2\mu}{m\omega_0^2}}, b(t)\sqrt{\frac{2\mu}{m\omega_0^2}}]$ there are two possible $\xi$ (mod $2\pi$) such that $x/(b(t)\sqrt{\frac{2\mu}{m\omega_0^2}}) = \cos\xi$. We write these two solutions as

$$\begin{cases} \xi(x, t) \equiv -\arccos\left(\frac{x}{b(t)}/\sqrt{\frac{2\mu}{m\omega_0^2}}\right) \in [-\pi, 0] \quad (\text{mod } 2\pi), \\ 2\pi - \xi(x, t) \in [0, \pi] \quad (\text{mod } 2\pi). \end{cases} \tag{37}$$

Then as in Sec. 2, observables that are sensitive only to left moving excitations will involve the chiral boson $\varphi(\xi - \tau)$ while the ones sensitive to right moving excitations will involve $\varphi(2\pi - \xi - \tau)$.

One can easily check that the metric (19) is given by $ds^2 = e^{2\sigma(x,t)}(-d\tau^2 + d\xi^2)$ with

$$e^{\sigma(x,t)} = \sqrt{\frac{2\mu}{m\omega_0^2}b(t)^2 - x^2}. \tag{38}$$

**Phases.** In the next sections we will be interested in correlation functions of observables which create/annihilate particles, i.e. observables that possess a non-zero U(1) charge. Thus we will have to be careful about phases. One phase that will appear is the WKB phase at the Fermi points $k_{F-}(x, t)$ and $k_{F+}(x, t)$. It is defined such that its differential is

$$d\phi_{\text{WKB}\mp}(x, t) = k_{F\mp}(x, t)dx - \frac{1}{\hbar}\left(\frac{\hbar^2 k_{F\mp}^2(x, t)}{2m} + V(x, t)\right)dt. \tag{39}$$

This has the following interpetation: if one imagines that one creates a free fermion at position $(x, k_{\mp}(x, t))$ in phase space, then the single-particle wavefunction of that new fermion will behave as $\exp(-i\frac{1}{\hbar}(\frac{\hbar^2 k_{F\mp}^2}{2m} + V)\delta t + i k_{F\mp}\delta x)$ at small distance $\delta x$ and small time $\delta t$; the latter phase then also multiplies the many-body wavefunction. Notice that the existence of a function $\phi_{\mathrm{WKB}\mp}$ satisfying Eq. (39) is slightly non-trivial: one must check that the cross derivatives are identical, and this turns out to be true because of the Burgers equation (12). Integrating Eq. (39), one gets (with $m = \omega_0 = \mu = 1$)

$$\phi_{\mathrm{WKB}\mp}(x, t) = \left(-\tau(t) + \frac{\dot{b}(t)}{b(t)}\frac{x^2}{2} \pm \left(\frac{x}{b(t)\sqrt{2}}\sqrt{1 - \left(\frac{x}{b(t)\sqrt{2}}\right)^2} - \arccos\frac{x}{b(t)\sqrt{2}}\right)\right)N, \quad (40)$$

up to an unimportant additive constant independent of $x$ and $t$.

Another phase that will show up is the combination of $\phi_{\mathrm{WKB}-}$ and $\phi_{\mathrm{WKB}+}$, corresponding to a local Galilean boost, such that

$$\partial_x \phi(x, t) = \frac{m}{\hbar} u(x, t).$$

This is satisfied by

$$\phi(x, t) \equiv \frac{1}{2}[\phi_{\mathrm{WKB}-}(x, t) + \phi_{\mathrm{WKB}+}(x, t)] = \left(-\tau(t) + \frac{\dot{b}(t)}{b(t)}\frac{x^2}{2}\right)N, \quad (41)$$

because $u = \frac{\hbar}{2m}[k_{F-} + k_{F+}]$.

## 4 Boson correlation functions

We now apply the above formalism to calculate the $2n$-point correlation function of boson creation/annihilation operators,

$$g_n((x_1, t_1), \ldots, (x_n, t_n), (x'_1, t'_1), \ldots, (x'_n, t'_n))$$
$$\equiv \left\langle \Psi^\dagger(x_1, t_1)\ldots\Psi^\dagger(x_n, t_n)\Psi(x'_1, t'_1)\ldots\Psi(x'_n, t'_n) \right\rangle, \quad (42)$$

always in the limit $1/N \sim \hbar \to 0$. To the best of our knowledge, such results cannot be obtained by any other method. From the numerical side, there exist efficient algorithms to compute the one-particle density matrix $g_1((x, t), (x', t'))$ in the initial state ($t = t' = 0$) based on the lattice version of this problem [78, 79], which map to the continuum problem at low fillings [80]; the extension to the dynamical situation ($t, t' \neq 0$) after a generic evolution is non trivial. Analytically, on the other hand, we are aware only of one result by Forrester *et al.* [60] (see also Gangardt [81] and the related work [82] for the extension to the anyonic case) which gives the one-particle density matrix $g_1(x, x')$ in the initial state in the same limit $1/N \sim \hbar \to 0$. The formula we obtain below naturally coincides with this known result in that particular case (see also Ref. [39] by two of us, where this is explained in great detail). The methods of Refs. [60, 81], however, do not seem to be easily generalizable to arbitrary times or to higher point correlations. The fact that the approach we take here leads to such results in a relatively simple way is a clear demonstration of its power and efficiency.

In this section we set $\omega_0 = \mu = m = 1$.

## 4.1  Identification of $\Psi^\dagger$, $\Psi$ with CFT operators

In order to use the above formalism to calculate the correlation function (42), we need to be able to express correlations of $\Psi^\dagger$ and $\Psi$ in the microscopic model —the inhomogeneous Tonks-Girardeau gas— as correlators of properly identified operators in the CFT. This was done in detail in Ref. [39], but we briefly recall the result for the convenience of the reader. The guiding principle is that any local operator in the microscopic model can be expanded as a sum of operators in the low-energy theory, and the latter sum can be organized in increasing order of scaling dimension of the operators. The only restriction on terms in the sum is that they should have the same symmetries as the microscopic model. For $\Psi^\dagger$ and $\Psi$, the CFT operators appearing in the expansion should carry a U(1) charge $\pm 1$, so the expansion should start as [39]

$$
\begin{aligned}
\Psi^\dagger(x,t) &\propto\ : e^{-\frac{i}{2}(\varphi_-(x,t)-\varphi_+(x,t))} : + \text{ less relevant operators} \\
\Psi(x,t) &\propto\ : e^{\frac{i}{2}(\varphi_-(x,t)-\varphi_+(x,t))} : + \text{ less relevant operators,}
\end{aligned}
\tag{43}
$$

where $\varphi_-$ and $\varphi_+$ are the right- and left-moving parts of the height field $h = \varphi_- + \varphi_+$. As we have seen in Sec. 2.7, the latter can also be written in terms of a single chiral boson $\varphi_-(\xi - \tau) = \varphi(\xi - \tau)$ and $\varphi_+(\xi + \tau) = -\varphi(2\pi - \xi - \tau)$.

In this paper we will keep only the leading order in Eq. (43), but in principle higher order could be taken into account as well, giving rise to subleading corrections in the limit $1/N \sim \hbar \to 0$ [46]. Importantly, we need to fix the prefactor of that leading term in Eq. (43). Since $\Psi^\dagger$ is homogeneous to a length to the power $-1/2$, while the vertex operator $: e^{-\frac{i}{2}(\varphi_- - \varphi_+)} :$ has scaling dimension $1/4$, we see by dimensional analysis that the prefactor must scale as $\rho(x,t)^{1/4}$, because the inverse density $\rho^{-1}$ is the only local length scale in the problem. Additionally, because the operators carry a U(1) charge, they must be multiplied by the U(1) phase identified in Eq. (41). This gives

$$
\begin{aligned}
\Psi^\dagger(x,t) &= C\,\rho(x,t)^{\frac{1}{4}}\, e^{-i\phi(x,t)}\, : e^{-\frac{i}{2}(\varphi_-(x,t)-\varphi_+(x,t))} : \\
\Psi(x,t) &= C^*\,\rho(x,t)^{\frac{1}{4}}\, e^{i\phi(x,t)}\, : e^{\frac{i}{2}(\varphi_-(x,t)-\varphi_+(x,t))} :,
\end{aligned}
\tag{44}
$$

where $C$ is a complex dimensionless constant which depends neither on position nor on time. Because of the global U(1) invariance, the phase of $C$ is arbitrary and does not affect the correlation function (42). The amplitude $|C|$ can be fixed from exact results for the homogeneous Tonks-Girardeau gas based on the analysis of Toeplitz determinants [83–86], see also the discussion in Ref. [39],

$$
|C|^2 = \frac{G^4(3/2)}{\sqrt{2\pi}} \simeq 0.521409,
\tag{45}
$$

where $G(.)$ is Barnes' G-function.

## 4.2  The one-particle density matrix $g_1((x,t),(x',t'))$

Now that the boson creation/annihilation operators are identified with CFT operators, the $2n$-point correlation function is straightforwardly computed by following the strategy of section 2. We shall start with the simplest case $n = 1$ and work out all the details.

Injecting the identification (44), we get

$$
\begin{aligned}
g_1((x,t),(x',t')) &= \left\langle \Psi^\dagger(x,t)\Psi(x',t') \right\rangle \\
&= |C|^2 \left( e^{-i(\phi(x,t)-\phi(x',t'))} \right) \left( \rho(x,t)\rho(x',t') \right)^{\frac{1}{4}} \\
&\quad \times \left\langle : e^{-\frac{i}{2}(\varphi_-(x,t)-\varphi_+(x,t))} : : e^{\frac{i}{2}(\varphi_-(x',t')-\varphi_+(x',t'))} : \right\rangle.
\end{aligned}
\tag{46}
$$

Next, as we have seen in section 2 the correlator can be evaluated in the initial state by tracing back the isothermal coordinates to $t = 0$. This is done by using formula (27) which gives

$$
g_1((x,t),(x',t')) = |C|^2 \left( e^{-i(\phi(x,t)-\phi(x',t'))} \right) \left( \frac{\rho(x,t)\rho(x',t')}{e^{\sigma(x,t)}e^{\sigma(x',t')}} \right)^{\frac{1}{4}}
$$
$$
\times \left\langle : e^{-\frac{i}{2}(\varphi(\xi-\tau)+\varphi(2\pi-\xi-\tau))} : : e^{\frac{i}{2}(\varphi(\xi'-\tau')+\varphi(2\pi-\xi'-\tau'))} : \right\rangle, \quad (47)
$$

where we have traded the right- and left-moving parts of the height field for the single chiral boson $\varphi(\xi)$ as explained in Sec. 2.7. Then, the 2-point correlator is equivalent to a 4-point correlator that is easily computed using Wick's theorem and the propagator of the chiral boson (28),

$$
\left\langle : e^{-\frac{i}{2}(\varphi(\xi-\tau)+\varphi(2\pi-\xi-\tau))} : : e^{\frac{i}{2}(\varphi(\xi'-\tau')+\varphi(2\pi-\xi'-\tau'))} : \right\rangle
$$
$$
= \left\langle : e^{-\frac{i}{2}(\varphi(\xi-\tau))} : : e^{-\frac{i}{2}(\varphi(2\pi-\xi-\tau))} : : e^{\frac{i}{2}(\varphi(\xi'-\tau'))} : : e^{\frac{i}{2}(\varphi(2\pi-\xi'-\tau'))} : \right\rangle
$$
$$
= \left( 2\sin\frac{\xi-\tau+\xi+\tau-2\pi}{2} \right)^{\frac{1}{4}} \left( 2\sin\frac{\xi-\tau-\xi'+\tau'}{2} \right)^{-\frac{1}{4}}
$$
$$
\times \left( 2\sin\frac{\xi-\tau+\xi'+\tau'-2\pi}{2} \right)^{-\frac{1}{4}} \left( 2\sin\frac{2\pi-\xi-\tau-\xi'+\tau'}{2} \right)^{-\frac{1}{4}}
$$
$$
\times \left( 2\sin\frac{-\xi-\tau+\xi'+\tau'}{2} \right)^{-\frac{1}{4}} \left( 2\sin\frac{\xi'-\tau'-2\pi+\xi'+\tau'}{2} \right)^{\frac{1}{4}}. \quad (48)
$$

Using $\frac{e^{\sigma(x,t)}}{\rho(x,t)} = \hbar\pi b(t)^2$ to simplify the scaling factors and Eq. (41) for the phases $\phi(x,t)$ and $\phi(x',t')$, it gives

$$
g_1((x,t),(x',t')) = \frac{|C|^2}{\sqrt{2\pi\hbar}} \frac{e^{-i\left( \frac{\dot{b}(t)}{b(t)} \frac{(x^2-x'^2)}{2} - \frac{(\tau-\tau')}{4} \right)N}}{\sqrt{b(t)b(t')}}
$$
$$
\times \frac{(\sin\xi)^{\frac{1}{4}}}{\left( \sin\frac{(\xi-\xi')-(\tau-\tau')}{2} \right)^{\frac{1}{4}} \left( \sin\frac{(\xi-\xi')+(\tau-\tau')}{2} \right)^{\frac{1}{4}}}
$$
$$
\times \frac{(\sin\xi')^{\frac{1}{4}}}{\left( \sin\frac{(\xi+\xi')+(\tau-\tau')}{2} \right)^{\frac{1}{4}} \left( \sin\frac{(\xi+\xi')-(\tau-\tau')}{2} \right)^{\frac{1}{4}}}. \quad (49)
$$

 The latter can be explictly written in terms of the physical coordinates $(x,t)$ and $(x',t')$ by plugging the expressions for the the isothermal coordinates (37)

$$
\begin{cases}
\xi(x,t) \equiv 2\pi - \arccos\left( \frac{x}{b(t)} / \sqrt{\frac{2\mu}{m\omega_0^2}} \right) \in [-\pi, 0] \pmod{2\pi}, \\
2\pi - \xi(x,t) \in [0, \pi] \pmod{2\pi},
\end{cases}
$$

and the density (36)

$$
\rho(x,t) = \frac{1}{b(t)} \times \frac{m\omega_0}{\pi\hbar} \sqrt{\frac{2\mu}{m\omega_0^2} - \frac{x^2}{b(t)^2}}.
$$

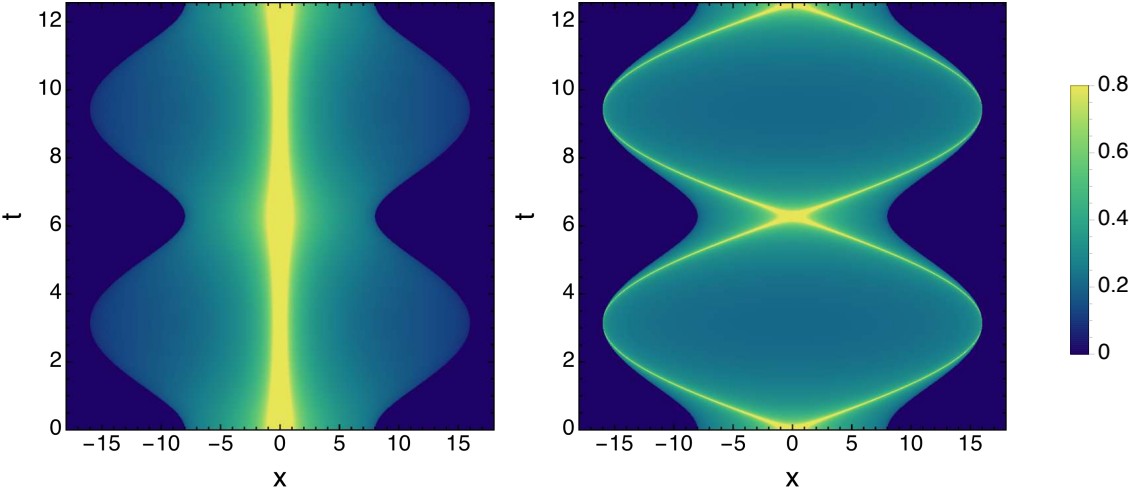

Figure 5: Absolute value of the one-particle density matrix at equal times, $|g_1(x,0;t)|$ (left) and at different times, $|g_1((x,t),(0,0))|$ (right), as a function of $x$ and $t$. The parameters are chosen as follows: $\omega_0 = 1, \omega_1 = 0.5, N = 32$.

Remarkably, the result at equal time $t = t'$ takes a nice and compact form,

$$
\begin{aligned}
g_1(x,x';t) \equiv g_1((x,t),(x',t)) &= \frac{|C|^2}{\sqrt{2\pi\hbar}} \frac{e^{-i\left(\frac{\dot{b}(t)}{b(t)}\frac{(x^2-x'^2)}{2}\right)N}}{b(t)} \frac{(\sin\xi)^{\frac{1}{4}}(\sin\xi')^{\frac{1}{4}}}{\left(\sin\frac{(\xi-\xi')}{2}\right)^{\frac{1}{2}}\left(\sin\frac{(\xi+\xi')}{2}\right)^{\frac{1}{2}}} \\
&= |C|^2 \frac{e^{-i\left(\frac{\dot{b}(t)}{b(t)}\frac{(x^2-x'^2)}{2}\right)N}}{\sqrt{b(t)}} \frac{\rho(x,t)^{\frac{1}{4}}\rho(x',t)^{\frac{1}{4}}}{\left|\frac{x-x'}{b(t)}\right|^{\frac{1}{2}}},
\end{aligned}
\tag{50}
$$

where the correlator in the first line precisely coincides with the one derived by two of us in [39]. In the second line, we see that the result obtained with our approach automatically satisfies the same scaling relation as the one found by Minguzzi and Gangardt [34]. It is also clear that at $t = 0$ (using $b(0) = 1$) we recover the result of Forrester et al. [60],

$$
g_1(x,x';0) = |C|^2 \frac{\rho(x,0)^{\frac{1}{4}}\rho(x',0)^{\frac{1}{4}}}{|x-x'|^{\frac{1}{2}}}.
\tag{51}
$$

In Fig. 5, we plot $g_1((x,t),(x',t'))$ for the particular case of a sudden quench of the frequency: $\omega(t) = \omega_0$ for $t \leq 0$ and $\omega(t) = \omega_1 \neq \omega_0$ for $t > 0$. In that case the scaling factor is $b(t) = \sqrt{1 + \left(\frac{\omega_0^2}{\omega_1^2} - 1\right)\sin^2(\omega_1 t)}$. We were not able to compare this result to a numerical evaluation of $g_1((x,t),(x',t'))$ in finite size, because we are not aware of a method that would allow to calculate that quantity at different times and for a large number of particles, even approximately. In contrast, the fact that our approach gives directly a closed analytic formula for the asymptotic behavior of that correlation function in the regime $1/N \sim \hbar \to 0$ shows that it is quite powerful.

### 4.3 The general $2n$-point function

The above derivation generalizes straightforwardly to the $2n$-point case. Following the same steps as for the one-particle density matrix, one finds

$$
\begin{aligned}
&g_n((x_1,t_1),\ldots,(x_n,t_n),(x'_1,t'_1),\ldots,(x'_n,t'_n))\\[4pt]
=\;&|C|^{2n}\left(\prod_{j=1}^{n}e^{-i(\phi(x_j,t_j)-\phi(x'_j,t'_j))}\right)\left(\prod_{i=1}^{n}\frac{\rho(x_i,t_i)\rho(x'_i,t'_i)}{e^{\sigma(x_i,t_i)}e^{\sigma(x'_i,t'_i)}}\right)^{\frac{1}{4}}\\[4pt]
&\times\left\langle\prod_{p=1}^{n}:e^{-\frac{i}{2}(\varphi_-(\xi_p-\tau_p)-\varphi_+(\xi_p+\tau_p))}:\prod_{q=1}^{n}:e^{\frac{i}{2}(\varphi_-(\xi'_q-\tau'_q)-\varphi_+(\xi'_q+\tau'_q))}:\right\rangle_{\text{flat}}\\[4pt]
=\;&\left(\frac{|C|^2}{\sqrt{\pi\hbar}}\right)^{n}\frac{e^{-i\sum_j(\phi(x_j,t_j)-\phi(x'_j,t'_j))}}{\sqrt{\prod_{i=1}^{n}b(t_i)b(t'_i)}}\\[4pt]
&\times\left\langle\prod_p:e^{-\frac{i}{2}\varphi(\xi_p-\tau_p)}:\prod_q:e^{\frac{i}{2}\varphi(2\pi-\xi'_q-\tau'_q)}:\prod_r:e^{-\frac{i}{2}\varphi(2\pi-\xi_r-\tau_r)}:\prod_s:e^{\frac{i}{2}\varphi(\xi'_s-\tau'_s)}:\right\rangle\\[4pt]
=\;&\left(\frac{|C|^2}{\sqrt{\pi\hbar}}\right)^{n}\frac{e^{-i\sum_j(\phi(x_j,t_j)-\phi(x'_j,t'_j))}}{\sqrt{\prod_{i=1}^{n}b(t_i)b(t'_i)}}\\[4pt]
&\times\left(\prod_{p_1<p_2}2\sin\frac{\xi_{p_1}-\tau_{p_1}-\xi_{p_2}+\tau_{p_2}}{2}\right)^{\frac{1}{4}}\left(\prod_{q_1<q_2}2\sin\frac{-\xi'_{q_1}-\tau'_{q_1}+\xi'_{q_2}+\tau'_{q_2}}{2}\right)^{\frac{1}{4}}\\[4pt]
&\times\left(\prod_{r_1<r_2}2\sin\frac{-\xi_{r_1}-\tau_{r_1}+\xi_{r_2}+\tau_{r_2}}{2}\right)^{\frac{1}{4}}\left(\prod_{s_1<s_2}2\sin\frac{\xi'_{s_1}-\tau'_{s_1}-\xi'_{s_2}+\tau'_{s_2}}{2}\right)^{\frac{1}{4}}\\[4pt]
&\times\left(\prod_{p,q}2\sin\frac{\xi_p-\tau_p+\xi'_q+\tau'_q-2\pi}{2}\right)^{-\frac{1}{4}}\left(\prod_{r,s}2\sin\frac{2\pi-\xi_r-\tau_r-\xi'_s+\tau'_s}{2}\right)^{-\frac{1}{4}}\\[4pt]
&\times\left(\prod_{p,r}2\sin\frac{\xi_p-\tau_p+\xi_r+\tau_r-2\pi}{2}\right)^{\frac{1}{4}}\left(\prod_{q,s}2\sin\frac{2\pi-\xi'_q-\tau'_q-\xi'_s+\tau'_s}{2}\right)^{\frac{1}{4}}\\[4pt]
&\times\left(\prod_{p,s}2\sin\frac{\xi_p-\tau_p-\xi'_s+\tau'_s}{2}\right)^{-\frac{1}{4}}\left(\prod_{q,r}2\sin\frac{-\xi'_q-\tau'_q+\xi_r+\tau_r}{2}\right)^{-\frac{1}{4}}.
\end{aligned}
\tag{52}
$$

We have used $\frac{e^{\sigma(x,t)}}{\rho(x,t)}=\hbar\pi b(t)^2$ to simplify the scaling factors, and Wick's theorem and the propagator of the chiral boson (28) to evaluate the correlator in the third line.

This is a rather complicated but fully explicit result. We stress that it is obtained relatively easily with our approach, as it boils down to simple calculations in a free boson CFT. As emphasized above, we do not know any other method that would allow to go that far in the calculation of correlation function for the trapped Tonks-Girardeau gas.

The result (52) can be put in a nicer form if one takes all the points at equal time, $t_1=\cdots=t_n=t'_1=\cdots=t'_n=t$. Indeed, after some algebra one arrives at the more compact formula

$$g_n(x_1, \ldots, x_n, x'_1, \ldots, x'_n; t)$$

$$= \left( \frac{|C|^2}{\sqrt{b(t)}} \right)^n e^{-i \sum_j (\phi(x_j, t) - \phi(x'_j, t))} \prod_i \rho(x_i, t)^{\frac{1}{4}} \prod_j \rho(x'_j, t)^{\frac{1}{4}}$$

$$\times \frac{\prod_{p_1 < p_2} \left| \frac{x_{p_1} - x_{p_2}}{b(t)} \right|^{\frac{1}{2}} \prod_{q_1 < q_2} \left| \frac{x'_{q_1} - x'_{q_2}}{b(t)} \right|^{\frac{1}{2}}}{\prod_{p,q} \left| \frac{x_p - x'_q}{b(t)} \right|^{\frac{1}{2}}}, \tag{53}$$

which simplifies even further in the initial state with $b(0) = 1$,

$$g_n(x_1, \ldots, x_n, x'_1, \ldots, x'_n; 0) = |C|^{2n} \prod_i \rho(x_i, 0)^{\frac{1}{4}} \prod_j \rho(x'_j, 0)^{\frac{1}{4}}$$

$$\times \frac{\prod_{p_1 < p_2} \left| x_{p_1} - x_{p_2} \right|^{\frac{1}{2}} \prod_{q_1 < q_2} \left| x'_{q_1} - x'_{q_2} \right|^{\frac{1}{2}}}{\prod_{p,q} \left| x_p - x'_q \right|^{\frac{1}{2}}}. \tag{54}$$

To the best of our knowledge, this formula is new and it generalizes the formula for $n = 1$ of Forrester et al. [60]. Also, one clearly sees that, in general, the $2n$-point function at equal time (53) satisfies the same scaling relation as the one found by Minguzzi and Gangardt in the $n = 1$ case [34].

## 5   The fermion propagator: prediction for the large-$N$ asymptotics and numerical check

To further illustrate the strategy outlined in section 2, we now compute the large-$N$ asymptotics of the fermion propagator at different times,

$$c(x, t, x', t') \equiv \langle \Psi_{\mathrm{F}}^\dagger(x, t) \Psi_{\mathrm{F}}(x', t') \rangle, \tag{55}$$

where $\Psi_{\mathrm{F}}^\dagger$, $\Psi_{\mathrm{F}}$ is the fermion creation/annihilation operator related to the bosonic one by Eq. (7). The final result for this propagator is given by formula (68) below. There are two contributions. The leading one is due to fermion excitations lying deep inside the Fermi sea, and is therefore beyond the approach sketched in section 2, strictly speaking. Nevertheless, we are able to obtain that term by elementary means. Then, the next leading contribution is due to low energy excitations and is the one in which we are truly interested, as it is the one which we can get from our approach. To explain this, we start from the homogeneous, translation-invariant case, where the two contributions are also present.

In this section we set $\mu = m = 1$.

### 5.1   Lessons from the infinite homogeneous case

We first look at the easier case of a free homogeneous Fermi gas on an infinite line. In that case the fermion propagator is

$$c(x, t, 0, 0) = \int_{-k_F}^{k_F} \frac{dk}{2\pi} e^{-ikx + i\hbar t \frac{k^2}{2}}, \tag{56}$$

where $k_F$ is the Fermi momentum (in terms of the notations of section 3, we have $k_{F-} = k_F$ and $k_{F+} = -k_F$). Of course, the integral could be evaluated exactly in terms of some error function. But instead, we are interested in evaluating its large $x$ and $t$ behavior by the stationary phase approximation. The approximation is valid in the limit of large $t$ and fixed ratio $x/t$, and it holds everywhere except along the lightcone $x/t = \pm v_F$, where $v_F = \hbar k_F$. There are two regimes to be considered: outside and inside the lightcone.

**Outside the lightcone ($|x| > v_F t$).** The integrand in Eq. (56) does not have a stationary point in the interval $[-k_F, k_F]$, therefore the main contribution to the integral in the stationary phase approximation comes from the regions around the two Fermi points $k \sim \pm k_F$,

$$
\begin{aligned}
c(x,t,0,0) & \simeq \int_{-\infty}^{k_F} \frac{dk}{2\pi} e^{-ik_F x + i\hbar t \frac{k_F^2}{2} - i(k-k_F)(x-v_F t)} + \int_{-k_F}^{\infty} \frac{dk}{2\pi} e^{ik_F x + i\hbar t \frac{k_F^2}{2} + i(k-k_F)(x+v_F t)} \\
& = \frac{i}{2\pi} \left[ \frac{e^{-ik_F x + i\hbar t \frac{k_F^2}{2}}}{(x - v_F t)} - \frac{e^{ik_F x + i\hbar t \frac{k_F^2}{2}}}{(x + v_F t)} \right].
\end{aligned}
\tag{57}
$$

**Inside the lightcone ($|x| < v_F t$).** There the phase of the integral (56) has a stationary point inside the region of integration (the point $\hbar k = x/t$). Therefore the main contribution comes from this stationary point, whereas the contributions from the endpoints of the interval $[-k_F, k_F]$ give the next to leading correction. Explicitly, we have

$$
\begin{aligned}
c(x,t,0,0) & \simeq \int_{-\infty}^{\infty} \frac{dk}{2\pi} e^{-i\frac{x^2}{2\hbar t} + i\hbar t \frac{(k-x/t)^2}{2}} \\
& \quad + \int_{-\infty}^{k_F} \frac{dk}{2\pi} e^{-ik_F x + i\hbar t \frac{k_F^2}{2} - i(k-k_F)(x-v_F t)} + \int_{-k_F}^{\infty} \frac{dk}{2\pi} e^{ik_F x + i\hbar t \frac{k_F^2}{2} + i(k-k_F)(x+v_F t)} \\
& = e^{i\frac{\pi}{4}} \left( \frac{1}{2\pi\hbar t} \right)^{1/2} e^{-i\frac{x^2}{2\hbar t}} + \frac{i}{2\pi} \left[ \frac{e^{-ik_F x + i\hbar t \frac{k_F^2}{2}}}{(x - v_F t)} - \frac{e^{ik_F x + i\hbar t \frac{k_F^2}{2}}}{(x + v_F t)} \right].
\end{aligned}
\tag{58}
$$

The leading term is thus associated with an excitation deep inside the Fermi sea, corresponding to a point inside the interval $[-k_F, k_F]$. The other two terms, however, clearly originate from the low-energy excitations around the Fermi points that are described by CFT.

We thus want to think of the fermion creation/annihilation operator as being a sum of three terms: an operator $d^\dagger/d$ exciting a state deep inside the Fermi sea, plus the right $(-)$ and left $(+)$ components of a Dirac field, or equivalently in terms of the chiral bosons $\varphi_-$ and $\varphi_+$, as vertex operators $: e^{-i\varphi_-} :$ and $: e^{i\varphi_+} :$,

$$
\Psi_{\mathrm{F}}^\dagger(x,t) = d^\dagger(x,t) + \frac{e^{i\frac{\pi}{4}}}{\sqrt{2\pi}} e^{-ik_F x + i\frac{k_F^2}{2}t} : e^{-i\varphi_-(x-k_F t)} : - \frac{e^{-i\frac{\pi}{4}}}{\sqrt{2\pi}} e^{ik_F x + i\frac{k_F^2}{2}t} : e^{i\varphi_+(x+k_F t)} :
\tag{59}
$$

$$
\Psi_{\mathrm{F}}(x,t) = d(x,t) + \frac{e^{i\frac{\pi}{4}}}{\sqrt{2\pi}} e^{ik_F x - i\frac{k_F^2}{2}t} : e^{i\varphi_-(x-k_F t)} : + \frac{e^{-i\frac{\pi}{4}}}{\sqrt{2\pi}} e^{-ik_F x - i\frac{k_F^2}{2}t} : e^{-i\varphi_+(x+k_F t)} :,
$$

then the CFT terms in Eqs. (57)-(58) correspond to the two-point functions in the free boson CFT on the infinite line,

$$
\left\langle : e^{-i\varphi_-(x-k_F t)} :: e^{i\varphi_-(0)} : \right\rangle = \frac{1}{x - k_F t}, \qquad \left\langle : e^{i\varphi_+(x+k_F t)} :: e^{-i\varphi_+(0)} : \right\rangle = \frac{1}{-x - k_F t}.
\tag{60}
$$

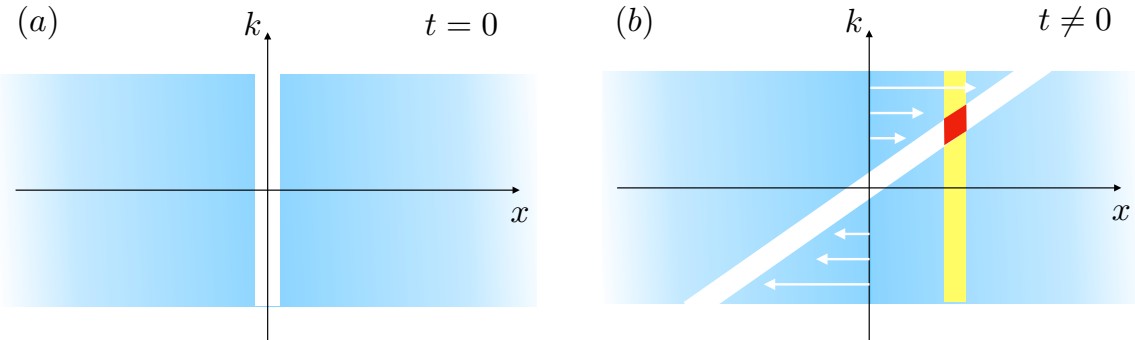

Figure 6: Wigner probability distribution for the homogeneous Fermi gas on an infinite line. Panel (a): we remove a particle at time $t = 0$ (white slice). Panel (b) The slice corresponding to the removed particle evolves. At time $t$ we want to create a particle in $x$ (yellow slice). The probability of doing it it proportional to the area shared by these two slides (red area).

The mixed terms involving $\varphi_-$ and $\varphi_+$ are zero on the infinite line, because there are no boundaries (there must be a boundary in order for left moving excitations to be reflected as right moving ones, thus inducing correlations between left and right sectors of the CFT). The term coming from the deeper excitation has the propagator

$$\left\langle d^\dagger(x,t)d(0,0)\right\rangle = \begin{cases} e^{i\frac{\pi}{4}}\left(\frac{1}{2\pi\hbar t}\right)^{1/2} e^{-i\frac{x^2}{2\hbar t}}, & \text{inside the lightcone} \\ 0, & \text{outside the lightcone,} \end{cases} \tag{61}$$

which has the following interpretation. The amplitude can be understood by looking at the Wigner function, which in this case is simply $n(x,k) = 1$ if $|k| < k_F$ and 0 otherwise. $d(0,0)$ destroys a particle at $x = 0$ and $t = 0$, corresponding in this language to removing a thin slice around $x = 0$, while $d^\dagger(x,t)$ creates a new one at position $x$ and time $t$. This is possible only inside the small area corresponding to the removed slice evolved up to time $t$ (see the red area in Figure 6). From the picture, one sees that such a probability decreases as $1/t$ and thus the corresponding amplitude goes as $1/\sqrt{t}$. The phase $e^{-i\frac{x^2}{2\hbar t}}$ is nothing but the semiclassical phase accumulated along the classical trajectory of the particle, namely

$$e^{-\frac{i}{\hbar}\int_0^t ds\left[\frac{1}{2}\dot{x}^2(s)-V(x(s))\right]}, \tag{62}$$

here with $x(s) = s\,x/t$ and $V(x) = 0$.

We conclude from this analysis of the homogeneous case that, while the approach outlined in section 2 is expected to give the correct $1/N \sim \hbar \to 0$ asymptotics for the fermion two-point function at different times, we will need to be careful about the contribution of excitations deep inside the Fermi sea, which is actually dominant. However the latter can be reconstructed easily by identifying the area $A(t)$ of a certain overlap in phase space (Figure 6), and the semiclassical phase (62).

## 5.2 Fermion propagator in the time-dependent harmonic trap

Now let us come back to the breathing gas in the time-dependent harmonic trap. To the best of our knowledge, exact results exist in the literature only for equal-time correlators [34,52–54]. Our method, instead, allows to obtain asymptotic results also at different times. Like in the

uniform case, we write the creation/annihilation operators in the microscopic model as a sum of three field, the first one acting deep inside the Fermi sea, and the other two being fields that excite low-energy excitations close to the Fermi points,

$$\Psi_F^\dagger(x,t) = d^\dagger(x,t) + \frac{e^{i\frac{\pi}{4}}}{\sqrt{2\pi}} e^{-i\phi_{WKB-}(x,t)} : e^{-i\varphi_-(\xi-\tau)} : - \frac{e^{-i\frac{\pi}{4}}}{\sqrt{2\pi}} e^{-i\phi_{WKB+}(x,t)} : e^{i\varphi_+(\xi+\tau)} :$$

(63)

$$\Psi_F(x,t) = d(x,t) + \frac{e^{i\frac{\pi}{4}}}{\sqrt{2\pi}} e^{i\phi_{WKB-}(x,t)} : e^{i\varphi_-(\xi-\tau)} : + \frac{e^{-i\frac{\pi}{4}}}{\sqrt{2\pi}} e^{i\phi_{WKB+}(x,t)} : e^{-i\varphi_+(\xi+\tau)} : .$$

The phases of the different terms are chosen in order to locally match the ones in the homogeneous case for small $|x-x'|$ and $|t-t'|$, see Eq. (59). The coordinate $\xi(x,t)$ is given by Eq. (37).

### 5.2.1 CFT contribution

We focus first on the contributions due to the vertex operators in (63), which is the one that is given by the approach outlined in section 2. After evaluating the two-point functions of the vertex operators, we arrive at

$$\frac{1}{2\pi} e^{-\frac{1}{2}\sigma(x,t)-\frac{1}{2}\sigma(x',t')} \left[ \frac{e^{-i[\phi_{WKB+}(x,t)-\phi_{WKB+}(x',t')]}}{2i\sin\frac{(\xi-\xi')+(\tau-\tau')}{2}} - \frac{e^{-i[\phi_{WKB-}(x,t)-\phi_{WKB-}(x',t')]}}{2i\sin\frac{(\xi-\xi')-(\tau+\tau')}{2}} \right.$$
$$\left. + \frac{e^{-i[\phi_{WKB+}(x,t)-\phi_{WKB-}(x',t')]}}{2\sin\frac{(\xi+\xi')+(\tau-\tau')}{2}} + \frac{e^{-i[\phi_{WKB-}(x,t)-\phi_{WKB+}(x',t')]}}{2\sin\frac{(\xi+\xi')-(\tau+\tau')}{2}} \right]. \quad (64)$$

The conformal factor $e^{\sigma(x,t)}$ is defined in Eq. (38).

### 5.2.2 Contribution from deep inside the Fermi sea

Like in the homogeneous case, there is a contribution coming from excitations deep inside the Fermi sea; we have learned that it consists of an amplitude which has a simple geometric interpretation, and of the semiclassical phase (62). We focus first on the amplitude.

**The amplitude.** It is proportional to the square root of the red area $A(t)$ shown in Figure 7. By virtue of the scaling approach, as time passes the evolution of the system can be viewed as a simple rescaling plus a rotation of phase space, encoded in the linear transformation sending a point $(x(0), k(0))$ to a point $(x(t), k(t))$ (see Sec. 3),

$$\begin{pmatrix} x(0)\sqrt{\omega_0} \\ \hbar k(0)/\sqrt{\omega_0} \end{pmatrix} \mapsto \begin{pmatrix} x(t)\sqrt{\omega_0} \\ \hbar k(t)/\sqrt{\omega_0} \end{pmatrix} = M(t) \begin{pmatrix} x(0)\sqrt{\omega_0} \\ \hbar k(0)/\sqrt{\omega_0} \end{pmatrix},$$

with

$$M(t) = \begin{pmatrix} b(t) & 0 \\ \dot{b}(t) & \frac{1}{b(t)} \end{pmatrix} \begin{pmatrix} \cos\tau(t) & \sin\tau(t) \\ -\sin\tau(t) & \cos\tau(t) \end{pmatrix}.$$

Then finding the area in red in Fig. 7 is an elementary geometric problem: for a white strip of initial width $\delta x$ the result is $\frac{\det M(t)}{[M(t)]_{12}}\omega_0\delta x^2$. Notice that when we go from $t'$ to $t$ we need to do that for $M(t)M(t')^{-1}$ instead of $M(t)$, so the area we are interested in is proportional to

$$A(t,t') = \frac{\omega_0 \det[M(t)M(t')^{-1}]}{[M(t)M(t')^{-1}]_{12}} = \frac{\omega_0}{b(t)b(t')\sin[\tau(t)-\tau(t')]}. \quad (65)$$

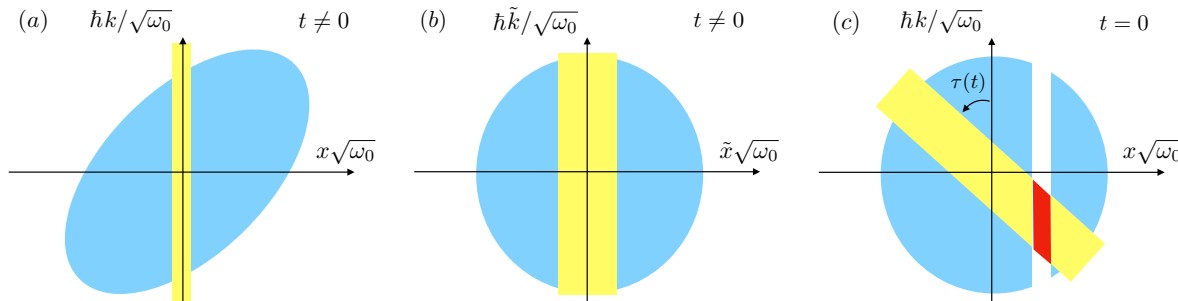

Figure 7: Wigner probability distribution in the harmonic trap after a generic harmonic protocol. Panel (a): The probability of having a particle at position $x = 0$ at $t \neq 0$ is proportional to the area of the yellow strip. Panel (b): Using the scaling approach, such strip gets a rescaled in the coordinates $\tilde{x} = x/b(t)$ and $\tilde{k} = (kb(t) + x\dot{b}(t))$. Note that in these variables the Wigner function is just a circle. Panel (c): In the original variables, the tilded coordinates just rotates, giving the rotated yellow strip. If at initial time we remove a particle at a given position $x \neq 0$, the probability of creating a new one at $x = 0$ at time $t$ is proportional to the red area.

**The phase.**   Like in the homogeneous case, the phase of $\langle d^{\dagger}(x,t)d(x',t') \rangle$ should be given by the minimum of the Lagrangian of a single classical particle, see Eq. (62). The classical trajectory of the particle is (see section 3) of the form $x(s) = \ell_0 b(s) \cos(\xi_0 + \tau(s))$ with $\ell_0$ and $\xi_0$ chosen such that $x(t) = x$ and $x(t') = x'$. This gives

$$x(s) = b(s) \frac{\frac{x}{b(t)} \sin(\tau(s) - \tau(t')) - \frac{x'}{b(t')} \sin(\tau(s) - \tau(t))}{\sin(\tau(t) - \tau(t'))}.$$

After some algebra, one finds that the phase given by the integral of the Lagrangian along that trajectory is

$$\exp\left[ -\frac{i}{\hbar} \int_t^{t'} \left( \frac{1}{2}(\dot{x}(s))^2 - V(x(s)) \right) ds \right]$$
$$= \exp\left[ -\frac{i}{2\hbar} \left( \frac{\omega_0}{\tan(\tau(t) - \tau(t'))} \left( \frac{x^2}{b(t)^2} + \frac{x'^2}{b(t')^2} \right) + \frac{x^2 \dot{b}(t)}{b(t)} \right. \right.$$
$$\left. \left. - \frac{x'^2 \dot{b}(t')}{b(t')} - \frac{2\omega_0}{\sin(\tau(t) - \tau(t'))} \frac{x}{b(t)} \frac{x'}{b(t')} \right) \right]. \quad (66)$$

Putting the phase and the amplitude together, we arrive at the conclusion that the contribution from excitations deep inside the Fermi sea takes the form

$$\langle d^{\dagger}(x,t)d(x',t') \rangle = e^{\frac{i\pi}{4}} \left( \frac{1}{2\pi\hbar} \frac{\omega_0}{b(t)b(t') \sin[\tau(t) - \tau(t')]} \right)^{\frac{1}{2}}$$
$$\times \exp\left[ -\frac{i}{2\hbar} \left( \frac{\omega_0}{\tan(\tau(t) - \tau(t'))} \left( \frac{x^2}{b(t)^2} + \frac{x'^2}{b(t')^2} \right) + \frac{x^2 \dot{b}(t)}{b(t)} \right. \right.$$
$$\left. \left. - \frac{x'^2 \dot{b}(t')}{b(t')} - \frac{2\omega_0}{\sin(\tau(t) - \tau(t'))} \frac{x}{b(t)} \frac{x'}{b(t')} \right) \right]. \quad (67)$$

### 5.2.3 Final result for $c(x, t, x', t')$ and comparison with numerics

We thus arrive at the following result for the asymptotics of the propagator, which is the sum of Eqs. (67) and (64). The asymptotics should be valid both inside and outside the lightcone (but not exactly along the lightcone, like in the infinite homogeneous case):

$$
c(x, t, x', t') = I(\xi, \xi', \tau - \tau') e^{\frac{i\pi}{4}} \left( \frac{A(t, t')}{2\pi} \right)^{\frac{1}{2}} \exp\left[ -i \int_t^{t'} \left( \frac{1}{2}(\partial_s x(s))^2 - V(x(s)) \right) ds \right]
$$
$$
+ \frac{1}{2\pi} e^{-\frac{1}{2}\sigma(x,t) - \frac{1}{2}\sigma(x',t')} \left[ \frac{e^{-i[\phi_{\mathrm{WKB}+}(x,t) - \phi_{\mathrm{WKB}+}(x',t')]}}{2i \sin \frac{(\xi - \xi') + (\tau - \tau')}{2}} - \frac{e^{-i[\phi_{\mathrm{WKB}-}(x,t) - \phi_{\mathrm{WKB}-}(x',t')]}}{2i \sin \frac{(\xi - \xi') - (\tau + \tau')}{2}} \right.
$$
$$
\left. + \frac{e^{-i[\phi_{\mathrm{WKB}+}(x,t) - \phi_{\mathrm{WKB}-}(x',t')]}}{2 \sin \frac{(\xi + \xi') + (\tau - \tau')}{2}} + \frac{e^{-i[\phi_{\mathrm{WKB}-}(x,t) - \phi_{\mathrm{WKB}+}(x',t')]}}{2 \sin \frac{(\xi + \xi') - (\tau + \tau')}{2}} \right]. \quad (68)
$$

Here $I(\xi, \xi', \tau)$ is the function which is one inside the lighcone and zero outside the lightcone, which is simple in the $(\xi, \tau)$ coordinates (see Figure 8): for $\tau \in [0, 2\pi]$ it can be written in terms of the Heaviside function $\Theta(.)$ as

$$
I(\xi, \xi', \tau) = \Theta(\tau - |\xi - \xi'|)\Theta(|\pi - \tau| - |\pi + \xi + \xi'|)\Theta(2\pi - \tau - |\xi - \xi'|) \quad (69)
$$

and then it is extended to other values of $\tau$ by periodicity: $I(\xi, \xi', \tau \pm 2\pi) = I(\xi, \xi', \tau)$.

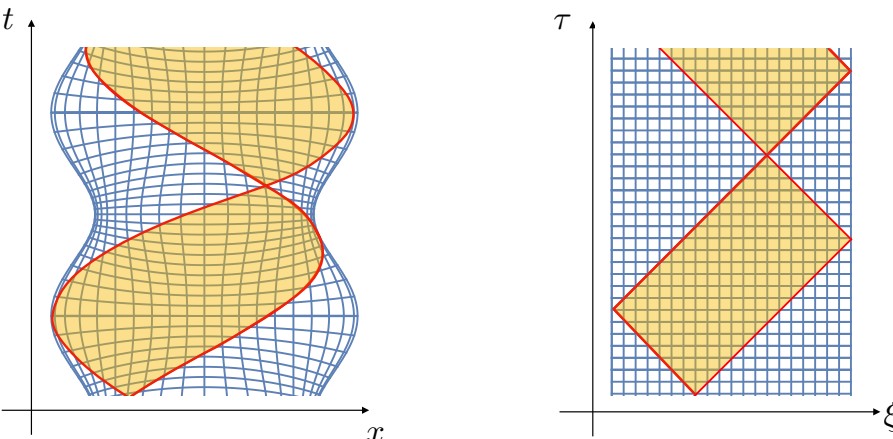

Figure 8: Left: Inside (yellow) and outside (white) lightcone regimes in the original $(x, t)$ coordinates. The continuous red line represents the lightcone. Right: Same in the $(\xi, \tau)$ coordinates. The grid shows the boundary of the system.

**Specialization to sudden quench of the frequency $\omega_0 \to \omega_1$ and comparison with numerics.** We now compare those formulas to numerical evaluation of the propagator at finite $N$ (see the Appendix), and to do this we specialize to the case of a sudden quench of the frequency: $\omega(t) = \omega_0$ if $t < 0$ and $\omega_1$ if $t > 0$. In that case $b(t) = \sqrt{1 + \left( \frac{\omega_0^2}{\omega_1^2} - 1 \right) \sin^2 \omega_1 t}$, and formula (67) for the contribution from deep inside the Fermi sea simplifies to

$$
e^{\frac{i\pi}{4}} \left( \frac{1}{2\pi\hbar} \frac{\omega_1}{\sin(\omega_1(t - t'))} \right)^{\frac{1}{2}} \exp\left[ -i \frac{\omega_1}{2\sin(\omega_1(t - t'))} \left[ (x^2 + x'^2)\cos(\omega_1(t - t')) - 2xx' \right] \right]. \quad (70)
$$

The comparison between the asymptotic formula (68) and the exact propagator at finite $N$, Eq. (73), is shown in Figure 9. The agreement is perfect both inside and outside the

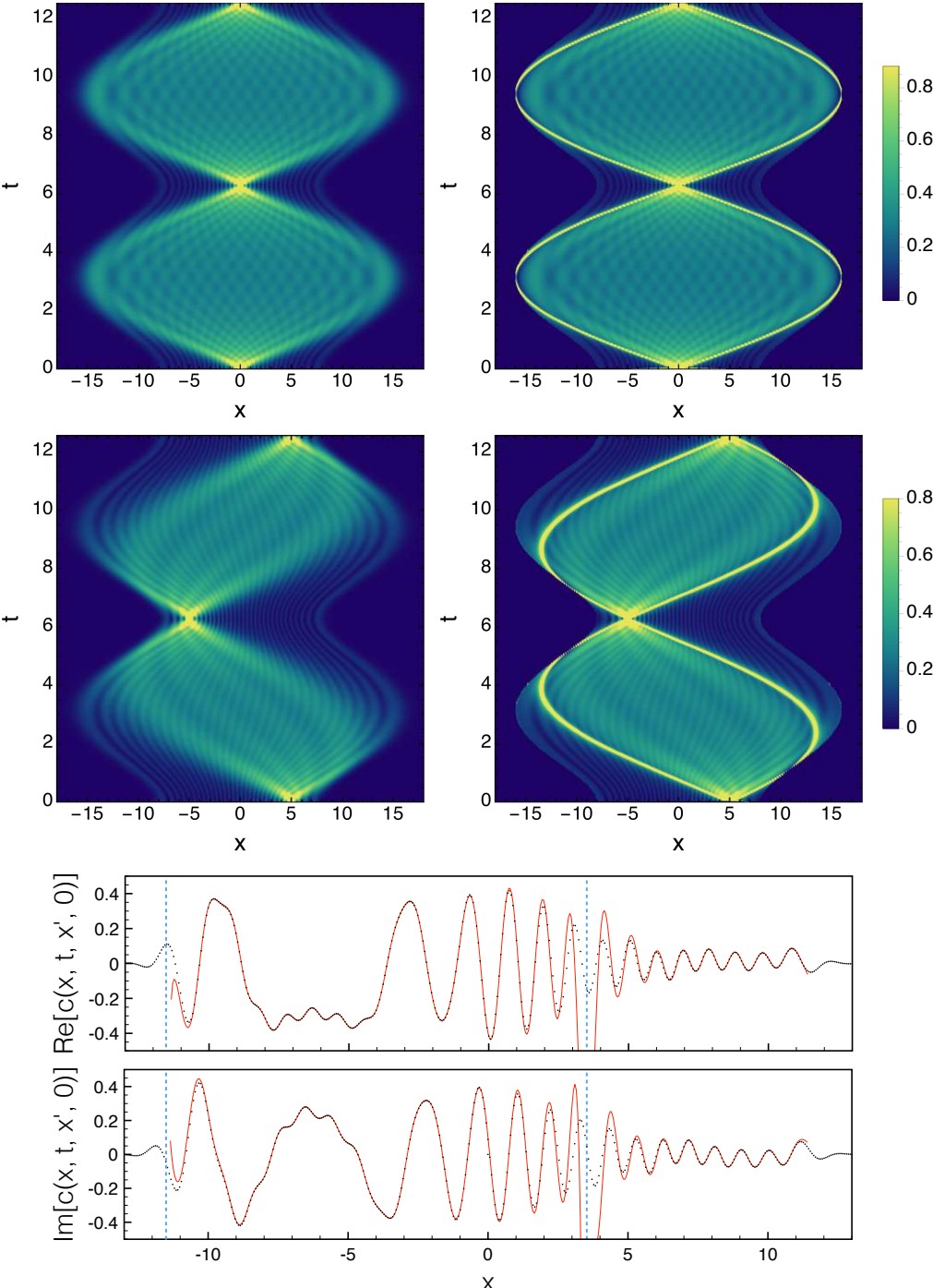

Figure 9: Fermionic propagator, $c(x, t, x', 0)$, Eq.(55), as a function of $x$ and $t$, for fixed values of $x'$. In the density plots the absolute value is shown. Upper row: $x' = 0$. Lower row: $x' = 5$. Left: Exact function from scaling approach, Eq. (73). Right: Our prediction, Eq. (68). The one dimensional plots below show the real and the imaginary part of $c(x, t, x', 0)$ as a function of $x$, with $x' = 5$ and fixed $t = 5$. The black dots are the exact function from scaling approach, Eq. (73). The red line is our prediction, Eq. (68). The dashed vertical lines are the positions of the lightcone. The parameters are chosen as follows: $\omega_0 = 1, \omega_1 = 0.5, N = 32$.

lightcone. However, one clearly sees a small region along the lightcone where our approach is not expected to give the correct result. Note also that at the edges of the system, some corrections are expected to occur [61, 87].

# 6 Conclusion

In this paper, we have made one step forward in the calculation of correlation functions of inhomogeneous one-dimensional critical systems (see e.g. Refs. [1, 37–51]), by considering a truly dynamical situation: a breathing gas of hard core bosons at zero temperature. In particular, we have found new formulas for the $1/N \sim \hbar \to 0$ asymptotics of $2n$-point functions of boson creation/annihilation operators, and also for fermionic observables for which we provided numerical checks.

This work could be extended in several directions. First, it would be interesting to study a case with multiple Fermi points, as shown in Fig. 1(b). This could give rise to interesting interference effects (as discussed for instance in Ref. [71]) or interesting behavior of the entanglement entropy —because the entanglement entropy should be sensitive to the number of Fermi points—. Another very interesting direction would be to study the problem of the breathing Lieb-Liniger gas at finite repulsion strength $g$, extending the techniques of Ref. [46] for the static case to a dynamical situation. It is likely that such a study would be mostly numerical, since already the classical hydrodynamic equations (14) would not be analytically solvable in that case. Nevertheless, it should be possible to express large scale correlation functions in terms of the Green's function of a certain generalized Laplacian (as in Ref. [46]), which would then lead to an interesting efficient numerical method for the calculation of correlation functions in that case.

The most exciting direction would perhaps be to use the new formula (53) found in this paper to investigate correlations of the momentum distribution of particles in the gas. Correlations of the momentum distributions are in principle measurable, for instance by time-of-flight. This has been done in the weakly interacting regime of the gas [88, 89], but, to our knowledge, not yet in the strongly interacting regime for which formula (53) would apply.

Finally, it would also be interesting to consider possible extension of our method to higher dimension, where even fewer results exist (see, for example, [90]).

# Acknowledgements

We are grateful to Fabian Essler for early collaboration and key discussions which motivated this paper, and to Pasquale Calabrese for many insightful comments and encouragements, and for collaboration on closely related projects. We thank Benjamin Doyon for many insightful discussions about quantum hydrodynamics and Generalized Hydrodynamics, and Stefano Scopa for discussions about the time-dependent harmonic oscillator and for pointing out Ref. [62]. We also thank Alvise Bastianello, Jean-Marie Stéphan, and Jacopo Viti for stimulating discussions and joint work on related projects.

PR would like to thank Tommaso De Lorenzo, Maurizio Fagotti, Nicolas Pavloff, Marcos Rigol, Shinsei Ryu, Hubert Saleur, Dam Thanh Son, Mikhail Svonarev, Denis Ullmo, and David Weiss for useful discussions.

PR and JD thank the Erwin Schrödinger Institut, Vienna, for hospitality during the program "Quantum Paths" in April 2018 and the Galileo Galilei Institute, Florence, for hospitality during the program "Entanglement in Quantum Systems" in June 2018. JD thanks SISSA for kind hospitality in September and October 2018.

This work was partially supported by the CNRS Mission Interdisciplinaire through the Défi Infiniti "MUSIQ" (YB, JD) and by ERC under Consolidator grant number 771536, NEMO (PR).

## A  Fermion propagator from the scaling approach

In Sec. 5, the numerical checks were performed using the exact fermion propagator obtained from the scaling approach (which was used, for instance, by Minguzzi and Gangardt [34]). Below, we simply summarize the main steps that lead to this exact result. Then, we will show that the equal-time asymptotics calculated from the latter formula correspond to the CFT contribution we computed with our approach.

In this section we set $\hbar = \omega_0 = m = 1$.

**Exact propagator.** Since we are dealing with a harmonic trap, the single particle wavefunctions at $t = 0$ are

$$\psi_n(x, 0) = \sqrt{\frac{(1/\pi)^{1/2}}{2^n n!}} H_n(x) e^{-\frac{x^2}{2}}, \tag{71}$$

where $H_n$ is the Hermite polynomial of order $n$. These are the eigenstates of the single-particle hamiltonian at $t = 0$ with energies $\varepsilon_n = n + \frac{1}{2}$. The scaling approach gives the time-evolved wavefunctions in terms of the scaling factor $b(t)$ which solves the Ermakov equation (33) and the function $\tau(t)$ defined in Eq. (34),

$$\psi_n(x, t) = \frac{1}{\sqrt{b(t)}} \psi_n\left(\frac{x}{b(t)}, 0\right) e^{i \frac{x^2}{2} \frac{\dot{b}(t)}{b(t)} - i\varepsilon_n \tau(t)}. \tag{72}$$

Then the propagator at different times ii the many-body ground state with $N$ particles is

$$\left\langle \Psi_F^\dagger(x, t) \Psi_F(x', t') \right\rangle = \sum_{n=0}^{N-1} \psi_n^*(x, t) \psi_n(x', t'). \tag{73}$$

This remarkably simple result follows from an elementary calculation. Let us call $c_n^\dagger(t) = \int dx\, \psi_n(x, t) \Psi_F^\dagger(x)$ the operator creating a fermion with wavefunction $\psi_n(x, t)$, then we see that the ground state at $t = 0$ is $\prod_{j=0}^{N-1} c_j^\dagger(0) |0\rangle$, and the state at time $t$ is

$$U(t, 0) \prod_{j=0}^{N-1} c_j^\dagger(0) |0\rangle = \prod_{j=0}^{N-1} \left( U(t, 0) c_j^\dagger(0) U^\dagger(t, 0) \right) |0\rangle = \prod_{j=0}^{N-1} c_j^\dagger(t) |0\rangle,$$

where $|0\rangle$ is the vacuum. and $U(t, t')$ is the evolution operator from $t'$ to $t > t'$:

$$U(t, t') \equiv \mathcal{T} \cdot \exp\left(-i \int_{t'}^{t} ds\, H(s)\right). \tag{74}$$

Then

$$
\begin{aligned}
U(t, t') \Psi_F(x') U(t', 0) \prod_j c_j^\dagger(0) |0\rangle &= U(t, t') \sum_{n'=0}^{N-1} \left( \psi_{n'}(x', t') \prod_{j \neq n'} c_j^\dagger(t') |0\rangle \right) \\
&= \sum_{n'=0}^{N-1} \left( \psi_{n'}(x', t') \prod_{j \neq n'} c_j^\dagger(t) |0\rangle \right),
\end{aligned}
$$

and

$$\langle 0| \prod_j c_j(0) U^\dagger(t,0) \Psi_F^\dagger(x) = \sum_{n=0}^{N-1} \left( \psi_n^*(x,t) \, \langle 0| \prod_{j \neq n} c_j(t) \right).$$

We get the propagator by taking the overlaps between these two states,

$$\left\langle \Psi_F^\dagger(x,t) \Psi_F(x',t') \right\rangle = \langle 0| \prod_j c_j(0) U^\dagger(t,0) \Psi_F^\dagger(x) U(t,t') \Psi_F(x') U(t',0) \prod_j c_j^\dagger(0) |0\rangle.$$

The non-zero terms are the ones with $n' = n$, and we get formula (73) for the propagator as claimed.

**Equal-time asymptotics.** At equal time $t' = t$, it is possible to evaluate the asymptotics of the propagator directly. In that case, the sum in (73) can be computed using the Christoffel-Darboux formula

$$\sum_{n=0}^{N-1} \frac{H_n(x) H_n(x')}{2^n n!} = \frac{H_{N-1}(x') H_N(x) - H_{N-1}(x) H_N(x')}{2^N (N-1)! (x - x')}.$$

If we let $N \to \infty$, we can inject the following asymptotics for the Hermite polynomials

$$e^{-\frac{x^2}{2}} \mathcal{H}_N(x) \sim \frac{2^{\frac{2N+1}{4}} \sqrt{N!}}{(\pi N)^{\frac{1}{4}}} \frac{1}{\sqrt{\sin(\alpha)}} \sin\left( \frac{2N+1}{4} (\sin(2\alpha) - 2\alpha) + \frac{3\pi}{4} \right),$$

where $x = \sqrt{2N+1} \cos(\alpha)$, with $\epsilon \leq \alpha \leq \pi - \epsilon$ and $\epsilon \to 0$ when $N \to \infty$. Prior to the complete formulation of our approach, we observed that the result could be put in the nice form

$$
\begin{aligned}
\left\langle \Psi_F^\dagger(x,t) \Psi_F(x',t) \right\rangle &= \frac{1}{2\pi} \frac{e^{-i[\phi(x,t) - \phi(x',t)]}}{b(t)} \left[ \left( 2N - \frac{x^2}{b(t)^2} \right) \left( 2N - \frac{x'^2}{b(t)^2} \right) \right]^{-\frac{1}{4}} \\
&\quad \times \left[ \frac{\sin\left( \phi^\star(x,t) - \phi^\star(x',t) \right)}{\sin\left( \frac{\xi(x,t) - \xi(x',t)}{2} \right)} + \frac{\cos\left( \phi^\star(x,t) + \phi^\star(x',t) \right)}{\sin\left( \frac{\xi(x,t) + \xi(x',t)}{2} \right)} \right],
\end{aligned}
\tag{75}
$$

where

$$\phi^\star(x,t) \equiv \frac{1}{2} \left[ \phi_{\mathrm{WKB}-}(x,t) - \phi_{\mathrm{WKB}+}(x,t) \right],$$

and $\xi(x,t)$, $\phi_{\mathrm{WKB}\mp}(x,t)$ and $\phi(x,t)$ are defined in Eqs. (37,40,41). What is remarkable here is that the resulting formula (75) corresponds exactly to the sum of the four terms given by the CFT contribution of section 5.2.1.

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
