# Peer review of "Conformal field theory on top of a breathing one-dimensional gas of hard core bosons"

_SciPost Physics, doi:SciPost Phys. 6, 051 (2019)_

## Round 2 · Referee Report · Olalla Castro-Alvaredo · 2019-3-12

Strengths

1) This paper tackles a very timely subject, namely that of the out-of-equilibrium dynamics of a many-body quantum system. In particular, a rather complicated situation is considered namely, that of a a gas of hard-core bosons trapped in a potential that varies periodically in time. This set-up is close to experimentally realizable situations.
2) Remarkably the authors find closed analytic formulae for many-point functions of local operators at different times in the thermodynamic limit. This is a rather unique situation which is probably quite particular for this model and choice of potential, but which still gives a powerful benchmark for future work.
3) The paper is well written and well presented. Despite being rather technical and many different techniques being used, it manages to present the results in an accessible and clear way.

Weaknesses

1) The main weakness is that the nice results obtained here seem only accessible in this form for this particular model, so the authors are treating a very simple and particular case and much of what they achieve here will not be accessible in other cases. At the same time, the authors are clear from the beginning about this and I believe their contribution is valuable as an example that stills captures features we may see in more general theories. So this is not a strong weakness.

2) There are also a few trivial typos that I report later.

Report

This is a good paper which solves a clearly defined, interesting, and timely problem, namely computing correlation functions of operators at different times in a many-body quantum system which is driven out of equilibrium. The out-of-equilibrium dynamics is generated by the presence of a potential (trap) which varies periodically (harmonically) in time. The original system (before setting up the trap) is a hard-core boson gas which may be seen as a particular limit of the Lieb-Liniger model (an integrable model).

The paper is well written and well structured. Many different techniques are used (from hydrodynamic equations to CFT in curved space-time) and these are explained with the level of detail that is required to make the paper understandable.

The results are rather impresive as they are very explicit. In particular closed expressions for many point functions at different times are presented (the likes of which I have not seen for any other example) and successfully tested against numerical simulations.

I think that the paper could be published in its current form as I have no strong criticism. There are a couple of small suggestions I propose in the next section, but some of these are optional and (in part) a matter of taste.

Requested changes

I noticed some small inconsistencies in the use of capitals for words derived from people's names. For instance "galilean" is written both as "galilean" and as "Galilean" in different parts of the paper. Similarly "lagrangian" and "Laplacian" are both used. I think both uses are perfectly fine, but the authors should try to be consistent throughout the paper.

There are also a couple of equations that are too long (e.g. (67)) so should consider their presentation.

Finally, I have a tiny suggestion. Even though the paper is well written I have personally found it a bit difficult to follow at times, as different techniques are introduced at each stage, interrupting the flow of the presentation of original results. I personally find it quite useful in long papers to have a section after the introduction called something like "Summary of the main results" where you can report the main formulae (say for instance (52)) and give an indication of the techniques that are required to obtain them. I know some sort of summary is already given at the end of Section 1 but I think you could have an expanded version of that just before section 2.

  • validity: top
  • significance: high
  • originality: high
  • clarity: top
  • formatting: good
  • grammar: excellent

Author:  Yannis Brun  on 2019-04-04  [id 485]

(in reply to Report 1 by Olalla Castro-Alvaredo on 2019-03-12)

We are grateful to the referee for their thorough report and very positive feedback on the manuscript. Below we address the remarks that appear in the report.

1) "I noticed some small inconsistencies in the use of capitals for words derived from people's names. For instance "galilean" is written both as "galilean" and as "Galilean" in different parts of the paper. Similarly "lagrangian" and "Laplacian" are both used. I think both uses are perfectly fine, but the authors should try to be consistent throughout the paper."

Thank you, we have fixed this.

2) "There are also a couple of equations that are too long (e.g. (67)) so should consider their presentation."

We fixed this.

3) "Finally, I have a tiny suggestion. Even though the paper is well written I have personally found it a bit difficult to follow at times, as different techniques are introduced at each stage, interrupting the flow of the presentation of original results. I personally find it quite useful in long papers to have a section after the introduction called something like "Summary of the main results" where you can report the main formulae (say for instance (52)) and give an indication of the techniques that are required to obtain them. I know some sort of summary is already given at the end of Section 1 but I think you could have an expanded version of that just before section 2."

We had actually written a first version of the draft with a subsection at the end of the introduction summarizing the main results of the paper, especially formulas (49,52,68). But this turned out to be a bad idea: to summarize the results for correlation functions, we had to introduce many notations early on in the introduction, so the introduction was very lengthy and painful to read. We had to reorganize our manuscript to make it more readable, and this is the version we submitted to Scipost.

So we thank the referee for this thoughtful suggestion, but unfortunately we do not think it can work (since we have already tried it, unsuccessfully). We think the best way to present this work is to put each result in the corresponding Section.

Anonymous on 2019-04-10  [id 494]

(in reply to Yannis Brun on 2019-04-04 [id 485])
Category:
remark

I thank the authors for their reply to my few comments. What I proposed was very minor and I am happy the authors considered it carefully nonetheless. I am happy for the paper to be published in its current form.

---

## Round 2 · Referee Report · Anonymous · 2019-3-13

Strengths

- Explicit and nice results
- Timely topic

Weaknesses

- just few details left to clarify

Report

This is a very nice paper with explicit and timely results. The authors take the classical hydrodynamics solution for the time evolution of density and velocity of a strongly interacting 1d quantum gas and on top of this develop a space and time dependent CFT to describe its fluctuations. I just have few question that I ask the authors to clarify in the paper:
-- how much their approach is different from local density approximation? Namely could they take simply the density and the momentum in eq 13,14 and from there write a CFT with the x,t -dependent geometry or there is more to this?
-- The authors focus on the correlation functions at the same time. Indeed dynamical correlation functions cannot be obtained by simple CFT and one would need something like non-linear Luttinger liquid. It would be interesting to know if their results in 4.2 for correlations at different times - that they obtain only for free fermion system - can be computed with a non linear Luttiger on top of the breathing cloud.
-- maybe in the introduction it would be good to stress that hydrodynamics well predict one-point functions but fails to predict two-point function (or more specifically, simply predicts them to be zero)

Requested changes

see report

  • validity: top
  • significance: high
  • originality: high
  • clarity: top
  • formatting: perfect
  • grammar: perfect

Author:  Yannis Brun  on 2019-04-04  [id 486]

(in reply to Report 2 on 2019-03-13)

We thank the referee for their positive comments on the manuscript. Below we answer the three questions raised in the report.

1) "How much their approach is different from local density approximation? Namely could they take simply the density and the momentum in eq 13,14 and from there write a CFT with the x,t -dependent geometry or there is more to this?"

In the static case, the approach is not different from the local density approximation (LDA). In fact, one can think of LDA just as "hydrostatics". To elaborate, in the hydrostatic case $u=0$, Eq.(14) simplifies to $ \partial_t \rho = 0 $ and $ \frac{1}{\rho} \partial_x \mathcal{P} = -\frac{1}{m} \partial_x V(x) $. Using the thermodynamic relation $ d\mathcal{P} = \rho s dT + \frac{\rho}{m} d \mu $, we get the constraint $ \partial_x [\mu + V(x)] =0 $. This constraint is obviously satisfied by $ \mu \mapsto \mu(x) = \mu - V(x) $, which is nothing but LDA. We added a discussion of this in the draft in section 2.2.

In the dynamical case ($u$ non-zero), we are indeed "simply taking the density and momentum of the Euler hydrodynamic equations and write a CFT with x,t-dependent geometry", as the referee says. This is explained in sections 2.3 and 2.4. There is not "more to this".

But getting explicit formulas for correlation functions is still a non-trivial task, and this is the reason why the paper is quite long.

2) "The authors focus on the correlation functions at the same time. Indeed dynamical correlation functions cannot be obtained by simple CFT and one would need something like non-linear Luttinger liquid. It would be interesting to know if their results in 4.2 for correlations at different times - that they obtain only for free fermion system - can be computed with a non linear Luttiger on top of the breathing cloud."

Here the referee is wrong. We do not focus solely on correlations at equal times. On the contrary, one of the main objectives of the paper is to go beyond the equal-time case (which would be accessible by other methods, like the scaling approach used by Minguzzi and Gangardt for this particular problem). This is clearly announced in the abstract and in the introduction in the last paragraph of section 1.1.

We stress that our main results, which are the explicit formulae in Eqs.(49,52,68) are valid for operators at different times.

Regarding the non-linear Luttinger liquid: the referee is right to say that it could be interesting to investigate these aspects. At the moment we do not know how to do this. In Sections 2, 3, and 4 of our paper, we focus exclusively in to the pure linear Luttinger liquid contribution.

However, we would like to point out to the referee that our Section 5 contains a detailed discussion of the contributions from "excitations deep inside the Fermi sea" in the context of the different-time fermion-fermion correlation $\left< \Psi^\dagger(x,t) \Psi (x',t') \right>$. This is clearly going in the direction of the non-linear Luttinger liquid.

3) "Maybe in the introduction it would be good to stress that hydrodynamics well predict one-point functions but fails to predict two-point function (or more specifically, simply predicts them to be zero)."

Thank you for that suggestion, we added this remark to the introduction. It is also mentioned in the last paragraph of Section 2.2.

---

## Round 3 · Author Response

Manuscript modified according to the referees recommendations, see answers to report 1 and 2.

---

## Round 3 · List of Changes

-upper-case letters in "Galilean", "Lagrangian", "Euclidean", etc. fixed.

-slight expansion of the introduction to emphasize that multi-point functions at equal time in classical hydrodynamics would be zero

-discussion of LDA/hydrostatics added in section 2.2

-reorganization of sections 2.4, 2.5, 2.6 in order to clarify some technical aspects about Jacobian factors in the calculation of CFT correlation functions

-formatting of long equations (for instance (68)) fixed

-some typos fixed

---

## Editorial Decision

published